# `PixelCAM`: Pixel Class Activation Mapping for Histology Image Classification and ROI Localization

**Alexis Guichemerre**[1]                                    ALEXIS.GUICHEMERRE.1@ENS.ETSMTL.CA
**Soufiane Belharbi**[1]                                       SOUFIANE.BELHARBI@ETSMTL.CA
**Mohammadhadi Shateri**[1]                           MOHAMMADHADI.SHATERI@ETSMTL.CA
**Luke McCaffrey**[2]                                          LUKE.MCCAFFREY@MCGILL.CA
**Eric Granger**[1]                                              ERIC.GRANGER@ETSMTL.CA

[1] *LIVIA, Dept. of Systems Engineering, ETS Montreal, Canada*

[2] *Goodman Cancer Research, Centre, Dept. of Oncology, McGill University, Canada*

**Editors:** Accepted for publication at MIDL 2025

## Abstract

Weakly supervised object localization (WSOL) methods allow training models to classify images and localize ROIs. WSOL only requires low-cost image-class annotations yet provides a visually interpretable classifier, which is important in histology image analysis. Standard WSOL methods rely on class activation mapping (CAM) methods to produce spatial localization maps according to a single- or two-step strategy. While both strategies have made significant progress, they still face several limitations with histology images. Single-step methods can easily result in under- or over-activation due to the limited visual ROI saliency in histology images and scarce localization cues. They also face the well-known issue of asynchronous convergence between classification and localization tasks. The two-step approach is sub-optimal because it is constrained to a frozen classifier, limiting the capacity for localization. Moreover, these methods also struggle when applied to out-of-distribution (OOD) datasets. In this paper, a multi-task approach for WSOL is introduced for simultaneous training of both tasks to address the asynchronous convergence problem. In particular, localization is performed in the pixel-feature space of an image encoder that is shared with classification. This allows learning discriminant features and accurate delineation of foreground/background regions to support ROI localization and image classification. We propose `PixelCAM`, a cost-effective foreground/background pixel-wise classifier in the pixel-feature space that allows for spatial object localization. Using partial-cross entropy, `PixelCAM` is trained using pixel pseudo-labels collected from a pretrained WSOL model. Both image and pixel-wise classifiers are trained simultaneously using standard gradient descent. In addition, our pixel classifier can easily be integrated into CNN- and transformer-based architectures without any modifications. Our extensive experiments[1] on `GlaS` and `CAMELYON16` cancer datasets show that `PixelCAM` can improve classification and localization performance when integrated with different WSOL methods. Most importantly, it provides robustness on both tasks for OOD data linked to different cancer types, with large domain shifts between training and testing image data.

**Keywords:** Deep Learning, Image Classification, Visual Interpretability, Weakly Supervised Object Localization, Histology Images, Out-Of-Distribution Data.

---

1. https://github.com/AlexisGuichemerreCode/PixelCAM

## 1. Introduction

Histology image analysis remains the gold standard for diagnosing cancers of the brain (Khalsa et al., 2020), breast (Veta et al., 2014), and colon (Xu et al., 2020). However, training deep learning (DL) models for accurate localization of cancerous regions of interest (ROIs) requires pixel-wise annotation by pathologists which is a complex and time-consuming task, especially over Whole Slide Images (WSI). WSOL has emerged as a low-cost training approach (Zhou, 2017). Using only image class supervision, a DL model can be trained to classify an image, but also to provide spatial localization of ROIs associated with that class (Belharbi et al., 2022a; Choe et al., 2020; Murtaza et al., 2024; Rony et al., 2023). This significantly reduces the large cost of annotation and the need for dense pixel supervision. In addition, it builds interpretable classifiers (Belharbi et al., 2022b; Neto et al., 2024) which is critical in medical domains such as in histology image analysis.

Recent progress in WSOL is dominated by CAM methods (Rony et al., 2023). Several methods extract spatial activation maps in a single step (Ilse et al., 2018; Oquab et al., 2015; Wu et al., 2023), and require a module on top of a feature extractor to construct these maps. Then, a spatial pooling module extracts per-class scores used for supervised training using image-class labels. However, without any explicit guidance at the pixel level, these CAM methods can lead to poor maps due to the challenging nature of histology images where objects are often not salient (Rony et al., 2023). This can lead to under- or over-activation which creates a high false negative/positive rate at the pixel level. Moreover, training a single model may face the issue of asynchronous convergence of classification and localization tasks (Choe et al., 2020; Murtaza et al., 2024; Rony et al., 2023).

To bypass this convergence issue and the lack of pixel guidance, a recent direction has emerged in WSOL where it aims at providing explicit localization cues as pseudo-labels with dual models (Belharbi et al., 2022a,c; Murtaza et al., 2023a; Wei et al., 2021; Zhang et al., 2020; Murtaza et al., 2025, 2022; Zhao et al., 2023; Murtaza et al., 2023b). In particular, a per-task model is considered where the localizer is trained using pseudo-labels (Zhao et al., 2023). This leads to more parameters and training cycles. In addition, both tasks are disconnected leading to unrelated decisions in both models. A different approach (Belharbi et al., 2022a) uses a single model where a decoder is combined with a frozen classifier. While this yields good results, this architecture is limited as it is tied to frozen features at many layers using skip connections. This prevents the localizer from a better adaptation and hinders its performance.

In addition, recent work has revealed a major limitation to WSOL methods when dealing with domain shift in histology analysis (Guichemerre et al., 2024). The performance of WSOL methods is shown to decline on both tasks with out-of-distribution (OOD), limiting their real-world application. Further inspection suggests that features at the pixel level can be the root cause. Since they lack direct localization supervision, a feature encoder may provide poorly separated features concerning classes, making them vulnerable to domain shift, and class confusion.

To alleviate the aforementioned issues, we propose a novel multi-task WSOL method for histology image analysis. It is based on called `PixelCAM`– a cost-effective foreground / background (FG/BG) pixel-wise classifier working in the pixel-feature space, allowing for spatial object localization. It aims to explicitly learn discriminant pixel-feature representa-

tions and accurate delineation of FG and BG regions. This improves the ROI localization and image classification accuracy. Training is achieved through localization pseudo-labels extracted from a pretrained WSOL model. In addition, such pixel-wise classification provides the model with robustness to OOD data. Our single-step multi-task framework allows to simultaneously perform classification and localization tasks. Multi-task training is therefore leveraged to learn rich features for both tasks, compared to the constrained learning of features in a two-step approach. Our approach cooperates to converge to a satisfying solution for both tasks. The multi-task optimization is performed using standard gradient descent by using image-class labels and pixel-wise pseudo-labels extracted from a pretrained WSOL model.

Our main contributions are summarized as follows. **(1)** A novel multi-task WSOL method called `PixelCAM` is proposed for histology image analysis. Multi-task optimization of both classification and localization tasks is achieved in a single step by integrating a pixel classifier at the pixel-feature level while leveraging localization pseudo-labels. `PixelCAM` alleviates the asynchronous convergence issue in WSOL, improves the performance of both tasks, and importantly, provides robustness to OOD data. Our pixel classifier is versatile in that it can easily be integrated into any CNN- or transform-based classifier architecture without modification. **(2)** We conduct extensive experiments on two public datasets for colon (`GlaS`), and breast cancer (`CAMELYON16`). Our method outperforms WSOL baseline methods on both tasks. Additionally, when dealing with large domain shifts across cancer types, our `PixelCAM` method can maintain a high level of accuracy, making it an ideal choice for OOD scenarios. We provide several ablations to further analyze our method.

## 2. Proposed Method

Let us denote by $\mathbb{D} = \{(\boldsymbol{X}, y)_i\}_{i=1}^N$ a training set with $N$ samples, where $\boldsymbol{X} : \Omega \subset \mathbb{R}^2$ is a 2d input image of dimension $h' \times w'$ and $y \in \{1, \cdots, K\}$ is its global label, with $K$ being the number of classes. Our model is composed of three parts (Fig.1): (a) feature extractor, (b) image classifier, and (c) pixel-wise classifier. (a) The feature extractor backbone $h$ with parameter $\boldsymbol{\theta}_1$ produces a tensor spatial features $h(\boldsymbol{X}) = \mathsf{F} \in \mathbb{R}^{h' \times w' \times d}$, with depth $d$. For simplicity, we consider that $\mathsf{F}$ has the same dimensions as the input image $\boldsymbol{X}$ through interpolation. (b) The global image classifier head $g$ classifies the image content, with parameters $\boldsymbol{\theta}_2$, where $g(\mathsf{F})$ is the per-class probability. (c) The pixel-wise[2] classifier $f$ which classifies the embedding $\mathsf{F}_{i,j,:} = \mathsf{F}_p \in \mathbb{R}^d$ of a pixel at location $(i, j, :)$ or simply $p$ in $\mathsf{F}$ with parameters $\boldsymbol{\theta}_3$ into either foreground (1) or background (0) classes. This is typically a linear classifier. We refer by $f(\mathsf{F}_p)_0$, $f(\mathsf{F}_p)_1$ as the probability for the pixel at location $p$ to be background and foreground, respectively. For simplicity, we use $f(\mathsf{F}_p)$ for pixel-class probability. Classifying all locations creates the two localization maps $\mathsf{S} = f(\mathsf{F}) \in [0, 1]^{h' \times w' \times 2}$, where $\mathsf{S}_{:,:,0}, \mathsf{S}_{:,:,1}$ or simply $\mathsf{S}^0, \mathsf{S}^1$ refer to the background and foreground maps, respectively. For simplicity, we refer by $y' \in \{0, 1\}$ as the pseudo-label of a pixel location $p$. Let us denote the standard cross-entropy by $\mathrm{H}(\cdot, \cdot)$. The total parameters of our model is referred to as $\boldsymbol{\theta}$

---

2. For simplicity, we refer to a location in a spatial tensor as a pixel. However, in DL models, such location typically covers more than one pixel in the image space due to the receptive field. Interpolation can be used to upscale the features to the full image size.

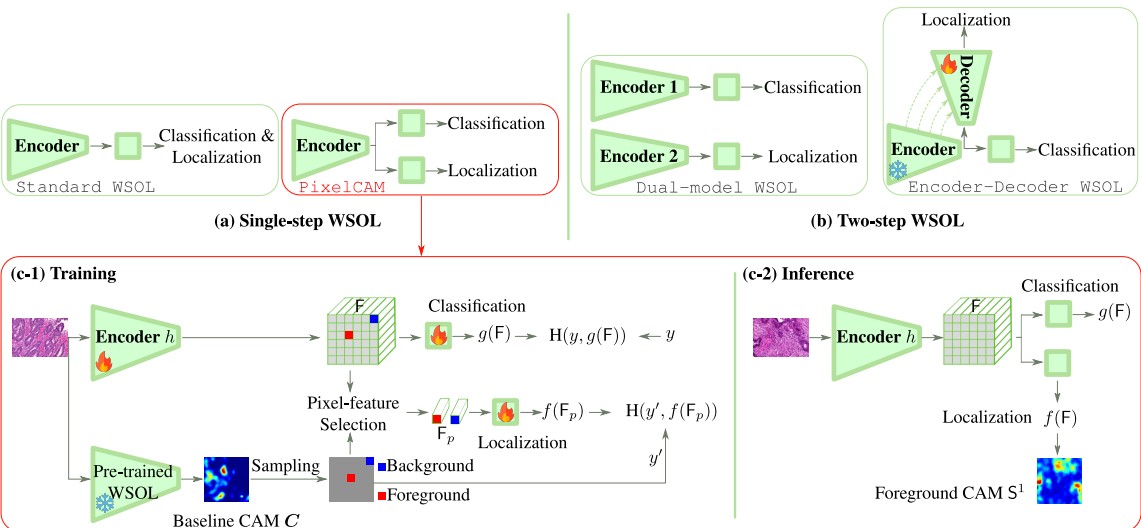

Figure 1: **Top row: WSOL learning strategies**. *(a) Single-step WSOL methods* perform classification and localization in one step. While standard WSOL uses a sequential approach using only image-class and lacks leveraging localization cues, `PixelCAM` relies on a multi-task training using both image-class and localization cues. *(b) Two-step WSOL methods* use a model per task or an encoder-decoder which are trained sequentially. **Bottom row: `PixelCAM` training (c-1) and inference (c-2)**. The training relies on pseudo-labels collected from a WSOL CAM classifier pretrained on the same dataset $\mathbb{D}$. The classification head aims to classify the extracted features $\mathsf{F}$. The pixel-wise classifier predicts the randomly selected locations as foreground or background. At test time, all locations are classified using the localization head $f$ producing a localization map, while classifying the whole image using the classification head $g$.

which is composed of $\{\boldsymbol{\theta}_1, \boldsymbol{\theta}_2, \boldsymbol{\theta}_3\}$. When $y, y'$ are used with H, it is the one-hot encoding version that is being considered.

To train the model for full image classification, we use standard cross-entropy over $g$:

$$\min_{\boldsymbol{\theta}_1, \boldsymbol{\theta}_2} \quad \mathsf{H}(y, g(\mathsf{F})) \ . \tag{1}$$

**Pixel-wise classifier**. Training the model using only Eq.1 builds a spatial feature $\mathsf{F}$ that is guided only by the image classification loss. It therefore lacks awareness of localization as it cannot directly access localization supervision cues. This may lead to poor pixel-wise features and confuse classes at the feature level. This is particularly true when dealing with less salient and strongly similar objects, as found in histology images. Subsequently, localization on top of these features is less reliable. Most importantly, introducing domain shift over the input image will lead to further feature confusion and failure by subsequent modules to localize and classify. To mitigate this issue, our pixel-classifier $f$ is used to create well-separated features between foreground/background in the pixel-feature space.

Training $f$ requires localization cues. Since the WSOL setup does not provide them, we resort to using a pretrained WSOL CAM classifier trained on the same train dataset $\mathbb{D}$, based on CNN or transformer, to acquire pseudo-labels. This approach is typically employed in WSOL to train a new DL model or a decoder for localization (Rony et al., 2023). However, this is cumbersome and requires another training phase with many additional parameters. To prevent this issue, our method trains a linear model to classify pixel embeddings while training the image classifier and sharing a full backbone. The number of parameters and training cycles is thereby reduced considerably.

To produce pixel-wise pseudo labels, standard WSOL methods (Rony et al., 2023) are adopted. We use the CAM $\boldsymbol{C}$ of the true image class $y$ from a WSOL CAM classifier pretrained on the same train dataset $\mathbb{D}$. Strong activations typically indicate foreground regions, while low activates point to background regions (Durand et al., 2017). Instead of directly fitting these regions, recent works showed that it is better to stochastically sample regions to avoid overfitting (Murtaza et al., 2023a; Belharbi et al., 2022c; Murtaza et al., 2023b). To sample $n$ random foreground pixel locations, a multinomial distribution is used with $\boldsymbol{C}$ as its pixel-sampling probability. Typically, this allows for sampling more frequently from high-activation locations. For background sampling, $1 - \boldsymbol{C}$ is instead used as a sampling probability. Sampled pixels are collected in the sets $\mathbb{C}^+, \mathbb{C}^-$ for foreground and background pixels, respectively. Note that sampled locations change for each image at every training step. This allows to explore different regions, and it reduces overfitting to specific regions. The sampled pseudo-labels can be directly used to train our pixel classifier, using partial cross-entropy since only part of the image space $\Omega$ is considered at once,

$$\min_{\boldsymbol{\theta}_1, \boldsymbol{\theta}_3} \quad \sum_{p \in \{\mathbb{C}^+ \cup \mathbb{C}^-\}} \mathrm{H}(y', f(\mathsf{F}_p)) \ . \tag{2}$$

The pixel pseudo-label $y'$ is collected based on the CAM of the true image class $y$. Therefore, $y'$ can be perceived as an instance of the object in that image. Our pixel-classifier learns to directly recognize the class $y$, allowing it to perform object localization.

**Total training loss**. Unlike previous works that use a two-step approach and train different models for classification and localization, `PixelCAM` trains a single model with two heads simultaneously. A multi-task optimization setup is considered to train for both tasks, with most parameters being shared. To this end, we use the following composite loss,

$$\min_{\boldsymbol{\theta}} \quad \mathrm{H}(y, g(\mathsf{F})) + \lambda \sum_{p \in \{\mathbb{C}^+ \cup \mathbb{C}^-\}} \mathrm{H}(y', f(\mathsf{F}_p)) \ , \tag{3}$$

where $\lambda$ is a balancing coefficient. Eq.3 is minimized using standard gradient descent to perform both tasks. This mitigates the convergence issue, and allows a single training cycle. It also ensures a cooperation between localization and classification to reach a better solution for both tasks at the same time. `PixelCAM` does not involve a considerable computation overhead. Given our lightweight pixel classifier, the number of additional parameters is negligible compared to standard DL models. Moreover, this pixel classifier is generic and can be integrated into CNN- or transformer-based models without architectural changes. Our pixel-classifier creates a boundary in the pixel feature space to well separate ROIs from noise and increase the feature's discriminant power. This helps localization, but also

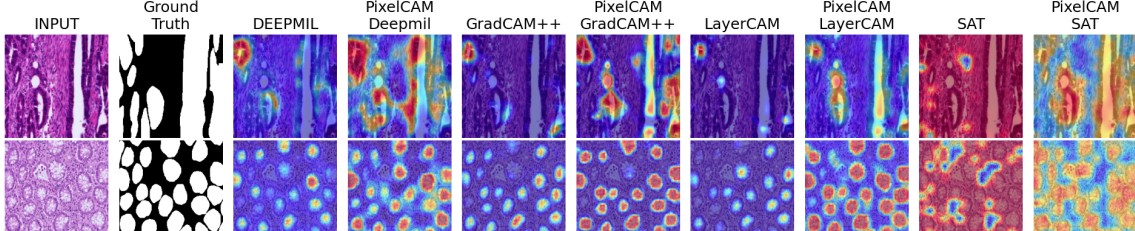

Figure 2: Standard WSOL setup. First column: input images from `GlaS`. Second column: Ground truth. Next columns: We display the visual CAM results for WSOL baseline without and with `PixelCAM`, respectively.

the subsequent classification module as they use the same spatial features for decision. Consequently, it makes features robust to OOD data, allowing better performance of both tasks in such scenario. In Section 3, we show that including this linear classifier into a standard deep WSOL model, allows us to improve its localization and classification accuracy. Moreover, it provides robustness to domain shift, a common issue in WSOL models for histology image analysis (Guichemerre et al., 2024).

## 3. Results and Discussion

### 3.1. Experimental Methodology[3]

*(a) Datasets.* Our experiments are performed on two standard public datasets for WSOL in histology images. In particular, we use `GlaS` dataset (Sirinukunwattana et al., 2015) for colon cancer, and the patch version of `CAMELYON16` (Ehteshami Bejnordi et al., 2017; Rony et al., 2023) for breast cancer. In both cases, we follow the same WSOL experimental protocol used in (Belharbi et al., 2022a; Guichemerre et al., 2024; Rony et al., 2023).

*(b) Implementation details.* To assess the impact of our method, we first train a baseline WSOL method alone, and then contrast its performance when combined with our method. The pseudo-labels are extracted from the baseline model. Different recent WSOL baseline methods are used from single- and tow-step families, including DeepMIL (Ilse et al., 2018), GradCAM++ (Chattopadhyay et al., 2018), LayerCAM (Jiang et al., 2021), SAT (Wu et al., 2023), and NEGEV (Belharbi et al., 2022a). For CNN-based models, we used ResNet-50 as the backbone, and for SAT, the transformer-based model DeiT-Tiny is used. In terms of performance measures, standard WSOL metrics are used, including image classification accuracy, and pixel-localization accuracy `PxAP` (Choe et al., 2020; Belharbi et al., 2022c,a; Guichemerre et al., 2024). In addition, pixel-wise true/false positive/negative rates are used as well. For the localization task, we compare all WSOL methods to the full-supervised case using the U-Net model.

---

3. A more detailed description of datasets and implementation are in Appendices B and C , respectively.

| | GlaS | | CAMELYON16 | |
|---|---|---|---|---|
| **WSOL models** | PxAP ↑ | CL ↑ | PxAP ↑ | CL ↑ |
| DeepMIL (Ilse et al., 2018) ICML † | 79.9 | **100.0** | 71.3 | 85.0 |
| DeepMIL w/ NEGEV (Belharbi et al., 2022a) MIDL ⋆ | **85.9** | **100.0** | 75.6 | **89.9** |
| DeepMIL w/ PixelCAM † | 85.5 | **100.0** | **75.7** | 88.2 |
| GradCAM++ (Chattopadhyay et al., 2018) WACV † | 76.8 | **100.0** | 49.1 | 63.4 |
| GradCAM++ w/ NEGEV (Belharbi et al., 2022a) MIDL ⋆ | 77.1 | **100.0** | **68.6** | **89.4** |
| GradCAM++ w/ PixelCAM † | **86.6** | **100.0** | 64.1 | 85.1 |
| LayerCAM (Jiang et al., 2021) IEEE TIP † | 75.1 | **100.0** | 33.2 | 84.8 |
| LayerCAM w/ NEGEV (Belharbi et al., 2022a) MIDL ⋆ | 73.8 | **100.0** | **66.8** | **89.1** |
| LayerCAM w/ PixelCAM † | **83.6** | **100.0** | 66.2 | **89.1** |
| SAT (Wu et al., 2023) ICCV † | 65.9 | 98.8 | 32.8 | 83.2 |
| SAT w/ PixelCAM † | **79.1** | **100.0** | **51.2** | **87.2** |
| U-Net (Ronneberger et al., 2015) MICCAI | 95.8 | n/a | 81.6 | n/a |

Table 1: Localization (PxAP) and classification (CL) accuracy on GlaS and CAMELYON16 test sets. † refers to model with a one-step approach while ⋆ refers to model from two-step family.

| | GlaS | | CAMELYON16 | |
|---|---|---|---|---|
| **WSOL models** | **T-stat ↑** | **p-value ↓** | **T-stat ↑** | **p-value ↓** |
| DeepMIL (Ilse et al., 2018) ICML † | 10.8 | $1.5 \times 10^{-11}$ | 4.8 | $4.1 \times 10^{-5}$ |
| GradCAM++ (Chattopadhyay et al., 2018) WACV † | 29.1 | $1.8 \times 10^{-22}$ | 17.9 | $7.4 \times 10^{-17}$ |
| LayerCAM (Jiang et al., 2021) IEEE TIP † | 12.7 | $4.0 \times 10^{-13}$ | 26.5 | $2.2 \times 10^{-21}$ |
| SAT (Wu et al., 2023) ICCV † | 58.3 | $8.9 \times 10^{-31}$ | 99.7 | $2.8 \times 10^{-37}$ |

Table 2: T-test statistics between baselines (DeepMIL, GradCAM++, LayerCAM, SAT) and our method (PixelCAM) for localization performance.

### 3.2. WSOL Results

We first evaluate PixelCAM using the standard WSOL protocol over GlaS and CAMELYON16 dataset (Rony et al., 2023). Results are reported in Tab. 1. Combining our method with a WSOL baseline often provides an improvement in localization accuracy. Over GlaS dataset, our method improves the PxAP performance by (+5.6%,+9.8%,+8.5%) compared to the WSOL baseline methods (DeepMIL, GradCAM++, LayerCAM) respectively, while on CAMELYON16 dataset, our method gains +4.4%,+15.0%,+33.0%, respectively. Fig. 2 shows visual prediction examples. The localization performance gains are supported by statistical significance, as confirmed by t-tests, in Tab. 2, yielding very low p-values (below 0.05). Additionally, our method improves classification performance as well. This indicates that well-separated classes at pixel-feature level in our model help improve localization but also facilitate global image classification. We further provide measures of how well are separated FG and BG pixel-features in Fig 3 on the GlaS test set. These results show that PixelCAM histogram of class separability index is shifted to the right compared to the baseline. This indicates a better class separability.

### 3.3. Out-of-Distribution Results

Additional experiments were conducted to assess the ability of PixelCAM to train robustness models on OOD data. To this end, consider the domain shift between GlaS (colon cancer)

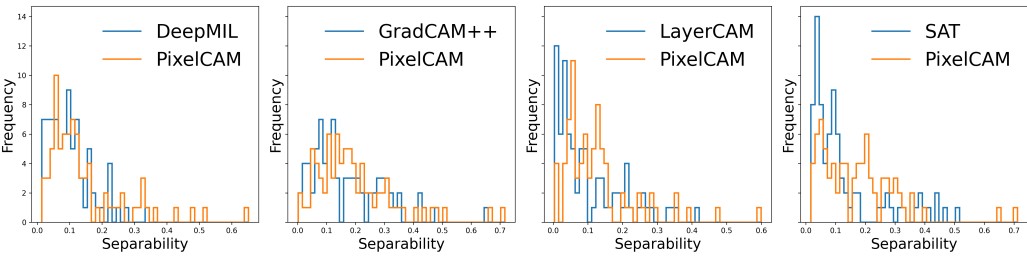

Figure 3: Histogram of foreground/background separability index for WSOL baseline methods alone vs when combined with `PixelCAM` on `GlaS` test set. The higher the value, the better the separability is. The definition of class separability is in Appendix D.

| | CAMELYON16 → GlaS | | GlaS → CAMELYON16 | |
|---|---|---|---|---|
| **WSOL models** | PxAP ↑ | CL ↑ | PxAP ↑ | CL ↑ |
| DeepMIL (Ilse et al., 2018) ICML † | 64.5 | 81.2 | 29.0 | **55.2** |
| DeepMIL w/ NEGEV (Belharbi et al., 2022a) MIDL ⋆ | 62.7 | 81.2 | 27.8 | 52.9 |
| DeepMIL w/ `PixelCAM` † | **69.1** | **83.8** | **30.2** | 52.5 |
| GradCAM++ (Chattopadhyay et al., 2018) WACV † | 52.9 | 53.7 | **39.1** | 52.4 |
| GradCAM++ w/ NEGEV (Belharbi et al., 2022a) MIDL ⋆ | **64.5** | **75.0** | 24.2 | 56.7 |
| GradCAM++ w/ `PixelCAM` † | 56.2 | 71.2 | 36.9 | **63.3** |
| LayerCAM (Jiang et al., 2021) IEEE TIP † | 59.5 | 77.5 | 30.4 | 51.4 |
| LayerCAM w/ NEGEV (Belharbi et al., 2022a) MIDL ⋆ | 63.9 | **83.7** | 27.9 | 52.3 |
| LayerCAM w/ `PixelCAM` † | **67.2** | 78.8 | **34.8** | **55.8** |
| SAT (Wu et al., 2023) ICCV † | 50.7 | 67.5 | **24.3** | 50.2 |
| SAT w/ `PixelCAM` † | **55.1** | **78.8** | 22.5 | **50.4** |

Table 3: OOD results: Localization (`PxAP`) and classification (`CL`) accuracy on target test dataset: `GlaS` and `CAMELYON16`. The symbol "†" refers to model with a one-step approach, while "⋆" refers to model from two-step family.

and `CAMELYON16` (breast cancer). This simulates a domain adaptation problem in the source-only case, where a model is pre-trained on source domain data, and then evaluated on target data. Results are reported in Tab. 3. As shown in (Guichemerre et al., 2024), WSOL methods typically decline when facing OOD in histology data. Overall, our `PixelCAM` method yields better performance over both tasks and across all baselines. This is mainly due to better separated pixel-features learned by our model which make them more robust to OOD data. This is demonstrated in the appendix D by inspecting the class separability of pixel-features of the target data. As presented in Tab. 3, using `PixelCAM` outperforms in general WSOL baselines alone on the case `CAMELYON16 → GlaS`. This improves `PxAP` scores by 4.6%, 3.3%, 7.7%, and 4.4% compared to DeepMIL, GradCAM++, LayerCAM, and SAT, respectively. Moreover, `PixelCAM` also improves image classification performance. The other case, `GlaS → CAMELYON16`, is much more challenging since the train source dataset is very small, compared to the very large and difficult target set. While the overall performance is low compared to when `GlaS` is the target, our method still yields better localization performance in general. Moreover, classification performance is improved overall across both scenarios. More results and interpretations are included in Appendices J and K.

We further evaluate the robustness of `PixelCAM` by altering the stainings in the test sets of `GlaS` using stain styles from the other dataset. The stainings were progressively

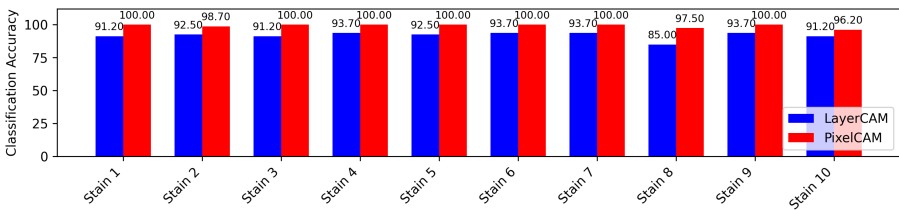

Figure 4: Classification (`CL`) accuracy on `GlaS` test sets with LayerCAM and `PixelCAM` with different stainings.

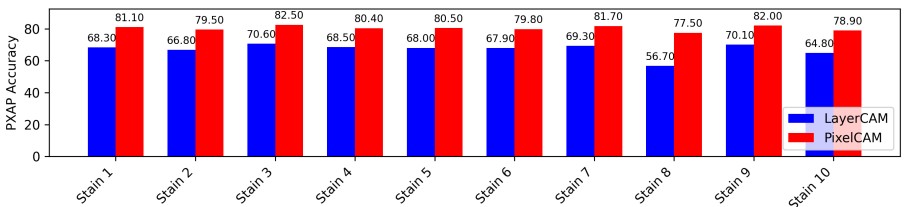

Figure 5: Localization (`PxAP`) accuracy on `GlaS` test sets with LayerCAM and `PixelCAM` with different stainings.

modified by selecting stain variations ordered by their distance to the original stain. Stain 1 introduces a minor alteration, while stain 10 represents a substantial shift. This setup simulates an increasing staining shift while keeping the organ type unchanged. Additional results for `CAMELYON16` are presented in Appendices H.

## 4. Conclusion

In this paper, `PixelCAM` is introduced for multi-task single-step WSOL method. It is based on a cost-effective FG/BG pixel-wise classifier in the pixel-feature space, allowing for spatial object localization. It leverages pixel-pseudo labels cues for localization. It aims at explicitly learning discriminant pixel-features and accurately separate FG/BG regions, promoting better ROI localization and image classification. Both tasks are optimized in parallel without incurring notable computational cost nor additional parameters. Our pixel classifier is versatile. It can easily be integrated into any CNN- or transform-based classifier architecture without modification. Results on two histology datasets showed that `PixelCAM` can improve WSOL baseline methods over both tasks, but also make them robust to OOD data. However, there is still room to improve our method by better optimizing both tasks while reducing their mutual negative impact. In addition, dealing with OOD in WSOL scenario is still an ongoing issue, and our method still requires more improvements despite its progress.

## Acknowledgments

This research was supported in part by the Canadian Institutes of Health Research, the Natural Sciences and Engineering Research Council of Canada, and the Digital Research Alliance of Canada.

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

This supplementary material contains the following points:

1. Related work

2. Datasets details

3. Training details for standard WSOL and OOD scenarios

4. Class separability index

5. More localization performance analysis

6. Additional results with different backbones

7. Computational complexity

8. Additional results for classification (`CL`) and localization (`PxAP`) accuracy on `CAMELYON16` test set with different stainings

9. Limitations and future work

10. Ablations:

    (a) Impact of `PixelCAM` depth
    (b) Impact of model selection of CAM pre-trained model
    (c) Impact of pixel sampling technique
    (d) Impact of the number of sampled pixels $n$
    (e) Impact of $\lambda$
    (f) Impact of using a pretrained WSOL model on same and external dataset for pseudo-labeling

11. More visualizations results:

    (a) Standard WSOL setup
    (b) Over Target set with OOD setup

## Appendix A. Related Work

WSOL has emerged as a low-cost training setup to localize objects while classifying an image content (Choe et al., 2020; Murtaza et al., 2024; Zhou, 2017). WSOL has also been extended to videos (Belharbi et al., 2023, 2025). Moreover, there has been a recent focus on designing WSOL methods for histology image analysis (Rony et al., 2023). This allows us to reduce the large annotation cost, in addition to building visually interpretable classifiers. **Single-step WSOL**. Several approaches address the WSOL problem in a single step (Choe et al., 2020; Murtaza et al., 2024; Rony et al., 2023) where a single model is trained to do both tasks, classification and localization. Usually, this is achieved by using a localization head followed by a spatial pooling layer to extract per-class probability. Early (Ilse et al., 2018; Selvaraju et al., 2020; Zhou et al., 2016) but also recent WSOL works (Wu et al., 2023;

Zhu et al., 2024; Gao et al., 2021) follow this strategy. Although this has brought significant improvements into the field, these models still face several limitations. Since only image-class labels are used as supervision without any localization cues, CAM localization can have poor performance, more so when dealing with less salient objects such as in histology images. Recent work (Rony et al., 2023) showed that without localization cues over histology images, these models often lead to highly unbalanced localization. This manifests either by under- or over-activations leading to high false negative/positive rates. In addition, this approach faces a great challenge in model selection as both tasks proceed in an asynchronous convergence (Choe et al., 2020; Murtaza et al., 2024; Rony et al., 2023). For instance, the model selected for best localization often yields poor classification.

**Two-step WSOL**. This strategy has emerged as a new line of research. In particular, a dual-model approach is used in combination with the usage of localization cues under the form of pseudo-labels (Murtaza et al., 2024; Rony et al., 2023). This provides direct guidance for the localization task, bypassing the issue of task convergence. Several works simply create a dedicated model per task: one for classification and another for localization (Wei et al., 2021; Zhang et al., 2020; Zhao et al., 2023). This leads to better performance since tasks are divided between two large models. However, this is a cumbersome approach as it drastically increases the number of parameters and training cycles. Most importantly, the localization CAMs are completely disconnected from the classification decision, making this strategy unreliable and less interpretable. A parallel direction overcomes this issue by using a decoder to act as a localizer on top of the classifier (Belharbi et al., 2022a,c; Murtaza et al., 2023a), in a similar way to a U-Net architecture (Ronneberger et al., 2015). This creates a direct relation between both tasks. However, the decoder has a limited learning capacity since it is tied to a frozen encoder at several layers via skip connections. This prevents the localizer from a better adaptation, and it limits its performance leading to a sub-optimal solution. In addition, it adds a significant number of parameters to the classifier.

**WSOL vs OOD**. In addition to these limitations, WSOL methods face challenges when dealing with domain shift which degrades the performance of both tasks (Guichemerre et al., 2024). Such shift is common in histology due to variations in stains, objects' structure, microscope type, and imaging centers. Further analysis shows that this issue could be rooted in the pixel features of the image encoder. Both classification and localization tasks depend heavily on these spatial features. Less flexible and poorly discriminant pixel features can lead to poor performance in both tasks since they both rely on these embeddings.

In summary, while existing WSOL methods have achieved great progress they still face several limitations. Single-step WSOL lacks the leverage of localization cues making them vulnerable to wrong localization when dealing with complex and less salient data such as histology images. This adds to the well known issue of asynchronous convergence of localization and classification tasks. On the other hand, two-step methods are cumbersome in terms of training cycles, and number of parameters. In addition, either both tasks are disconnected leading to misaligned decisions or the localization has a tied capacity due to frozen backbone. Our method comes as simple yet efficient alternative. It performs WSOL in a single-step within a multi-task framework where both tasks are optimized simultaneously. This leads to a single training cycle and facilitate convergence issue. It allows sharing parameters between both tasks making it parameter-efficient. But also, it

can leverage localization cues such as pseudo-labels. Finally, well separating features at pixel level makes our model robust to OOD data.

## Appendix B. Datasets

**GlaS.** The `GlaS` dataset is used for the diagnosis of colon cancer. The dataset consists of 165 images from 16 Hematoxylin and Eosin (H&E) and includes labels at both pixel-level and image-level (benign or malign). The dataset consists of 67 images for training, 18 for validation, and 80 for testing. We use the same protocol as in (Rony et al., 2023; Guichemerre et al., 2024; Belharbi et al., 2022a). Similarly, we use 3 samples per class with full supervision in validation set for model selection for localization (`B-LOC`).

**CAMELYON16.** A patch-based benchmark (Rony et al., 2023) is extracted from the `CAMELYON16` dataset that contains 399 Whole slide images categorized into two classes (normal and metastasic) for the detection of breast cancer metastases in H&E-stained tissue sections of sentinel lymph nodes. Patch extraction of size $512 \times 512$ follows a protocol established by (Rony et al., 2023; Guichemerre et al., 2024; Belharbi et al., 2022a) to obtain patches with annotations at the image and pixel level. The dataset contains a total of 48870 images, including 24348 for training, 8850 for validation, and 15664 for testing. From the validation dataset, 6 examples per class are randomly selected to be fully supervised to perform model selection for localization (`B-LOC`) similarly to (Rony et al., 2023; Guichemerre et al., 2024; Belharbi et al., 2022a).

## Appendix C. WSOL Training Details

For pretraining a WSOL baseline method on the data, we use the same setup as in (Guichemerre et al., 2024). The first part of the model training is defined using the backbone trained on ImageNet (Krizhevsky et al., 2012). The training is done using SGD with a batch size of 32 (Guichemerre et al., 2024). For the `GlaS` dataset, training is performed over 1000 epochs, and 20 epochs for `CAMELYON16`. A weight decay of $10^{-4}$ is also used. During training, images are resized to $256 \times 256$, then randomly cropped to $224 \times 224$. A hyperparameter search was conducted for the learning rate parameters among the values {0.0001, 0.001, 0.01}, and its decaying factor among {0.1, 0.4, 0.9} following (Guichemerre et al., 2024). In the second phase of training `PixelCAM`, we use the CAMs generated by the previous method and continue the training using the same setup as defined previously. For the OOD scenario, the source model is trained using the same setup on a source dataset and evaluated on a target dataset.

## Appendix D. Class Separability Index

In this section, we present the class separability index between foreground (FG) and background (BG) classes at pixel-feature level over test set of both datasets `GlaS` and `CAMELYON16`. To measure how well FG/BG classes are separated in the feature space of pixels, we resort to the class separability index (Duda et al., 2000). The class separability, $J$, is based on the Within-class scatter matrix ($\boldsymbol{S}_W$) and the Between-class scatter matrix ($\boldsymbol{S}_B$) of pixel-features, defined as follows,

$$S_W = \sum_{i=1}^{c} \left[ \sum_{j=1}^{n_i} (\boldsymbol{x}_{i,j} - \boldsymbol{m}_i)(\boldsymbol{x}_{i,j} - \boldsymbol{m}_i)^\top \right] , \tag{4}$$

$$S_B = \sum_{i=1}^{c} n_i (\boldsymbol{m}_i - \boldsymbol{m})(\boldsymbol{m}_i - \boldsymbol{m})^\top , \tag{5}$$

where $c$ is the number of class (in our case $c = 2$ for foreground and background). Note that this index $J$ is computed over a single image using the pixel-labels. Let $n_i$ $(i = 0, .., c)$ be the number of pixels in the $i$-th class. The feature vector $\boldsymbol{x}_{i,j} \in \mathbb{R}^d$ denotes the $j$-th pixel of the $i$-th class. It represents the vector $\mathsf{F}_p$ for the pixel $p$ in the main paper. The mean vector of the $i$-th class is given by $\boldsymbol{m_i}$, while $\boldsymbol{x}$ represents the mean vector computed over all feature vectors. The class separability denoted $J$ is defined as follows (Duda et al., 2000),

$$J = \frac{\mathrm{tr}(\boldsymbol{S}_B)}{\mathrm{tr}(\boldsymbol{S}_W)} . \tag{6}$$

In Table 4 and 5, we provide the average class separability over normal and cancer classes and the overview over the entire dataset. As we can observe, `PixelCAM` improves the class separability between FG and BG of WSOL baseline methods on both dataset except for GradCAM++ on `CAMELYON16`. This separability is explained by the high number of images with a low separability as illustrated in Fig 6. The improvement of `PixelCAM` in the OOD scenario compared to the WSOL baseline methods such as DeepMIL and LayerCAM on `GlaS` and `CAMELYON16` can be attributed to better separability as shown in Figs 7 and 8. We provide examples of features separability at the pixel level in section K.

| | GlaS | | | CAMELYON16 | | |
|---|---|---|---|---|---|---|
| **WSOL models** | Normal | Cancer | All | Normal | Cancer | All |
| DeepMil (Ilse et al., 2018) ICML | 0.13 | 0.07 | 0.10 | - | 0.11 | 0.11 |
| DeepMil w/ `PixelCAM` | **0.21** | **0.09** | **0.14** | - | **0.17** | **0.17** |
| GradCAM++ (Chattopadhyay et al., 2018) WACV | 0.22 | 0.13 | 0.17 | - | **0.22** | **0.22** |
| GradCAM++ w/ `PixelCAM` | **0.25** | **0.15** | **0.20** | - | 0.16 | 0.16 |
| LayerCAM (Jiang et al., 2021) IEEE TIP | **0.17** | 0.04 | 0.10 | - | 0.07 | 0.07 |
| LayerCAM w/ `PixelCAM` | 0.14 | **0.12** | **0.13** | - | **0.17** | **0.17** |
| SAT (Wu et al., 2023) ICCV | 0.21 | 0.06 | 0.13 | - | 0.17 | 0.17 |
| SAT w/ `PixelCAM` | **0.25** | **0.10** | **0.17** | - | **0.22** | **0.22** |

Table 4: Standard WSOL setup: Average class separability index $J$ between FG/BG pixel-features on test set `GlaS` and `CAMELYON16` for WSOL baselines with and without `PixelCAM`. The higher $J$ is, the more classes are separated.

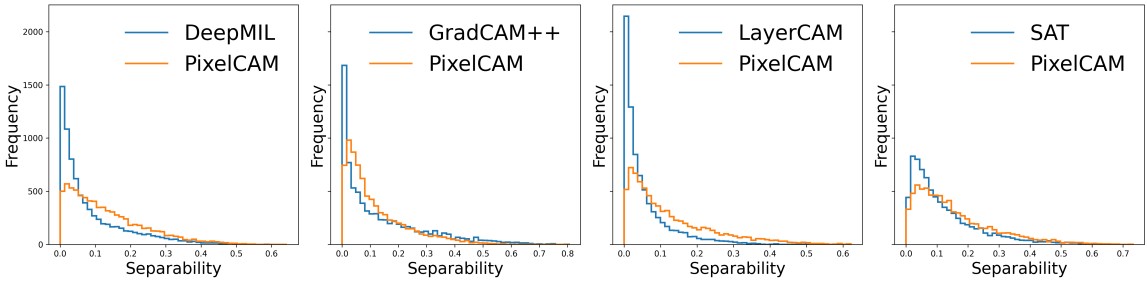

Figure 6: Standard WSOL setup: Histogram of class separability index $J$ between FG/BG pixel-features on test set `CAMELYON16` for WSOL baselines with and without `PixelCAM`. The higher $J$ is, the more classes are separated.

| WSOL models | CAMELYON16 $\rightarrow$ GlaS | | | GlaS $\rightarrow$ CAMELYON16 | | |
|---|---|---|---|---|---|---|
| | Normal | Cancer | All | Normal | Cancer | All |
| DeepMil (Ilse et al., 2018) ICML | 0.05 | 0.11 | 0.08 | - | **0.05** | **0.05** |
| DeepMil w/ `PixelCAM` | **0.06** | **0.13** | **0.10** | - | **0.05** | **0.05** |
| GradCAM++ (Chattopadhyay et al., 2018) WACV | **0.08** | **0.19** | **0.14** | - | **0.08** | **0.08** |
| GradCAM++ w/ `PixelCAM` | 0.05 | 0.09 | 0.07 | - | 0.07 | 0.07 |
| LayerCAM (Jiang et al., 2021) IEEE TIP | 0.02 | **0.13** | 0.08 | - | 0.04 | 0.04 |
| LayerCAM w/ `PixelCAM` | **0.06** | 0.11 | **0.09** | - | **0.09** | **0.09** |
| SAT (Wu et al., 2023) ICCV | 0.11 | 0.09 | 0.10 | - | 0.08 | 0.08 |
| SAT w/ `PixelCAM` | **0.12** | **0.12** | **0.12** | - | **0.09** | **0.09** |

Table 5: OOD setup: Average class separability index $J$ between FG/BG pixel-features on target test set `GlaS` and `CAMELYON16` for both OOD cases: `CAMELYON16` $\rightarrow$ `GlaS` and `GlaS` $\rightarrow$ `CAMELYON16` for WSOL baselines with and without `PixelCAM`. The higher $J$ is, the more classes are separated.

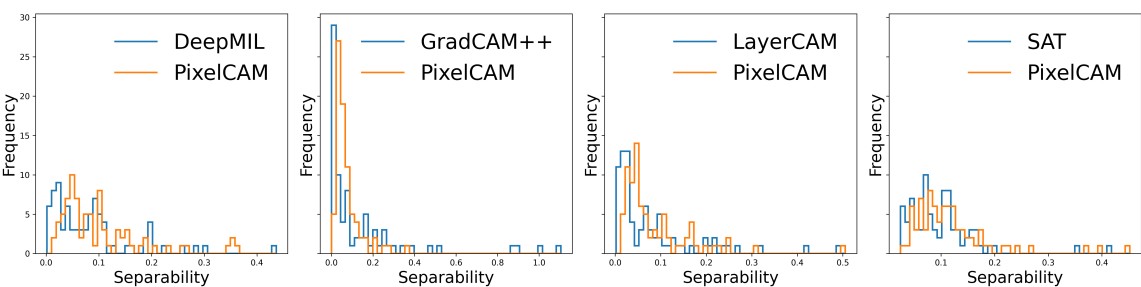

Figure 7: OOD setup: Histogram of class separability index $J$ between FG/BG pixel-features on target test set `GlaS` for the OOD case: `CAMELYON16` $\rightarrow$ `GlaS` for WSOL baselines with and without `PixelCAM`. The higher $J$ is, the more classes are separated.

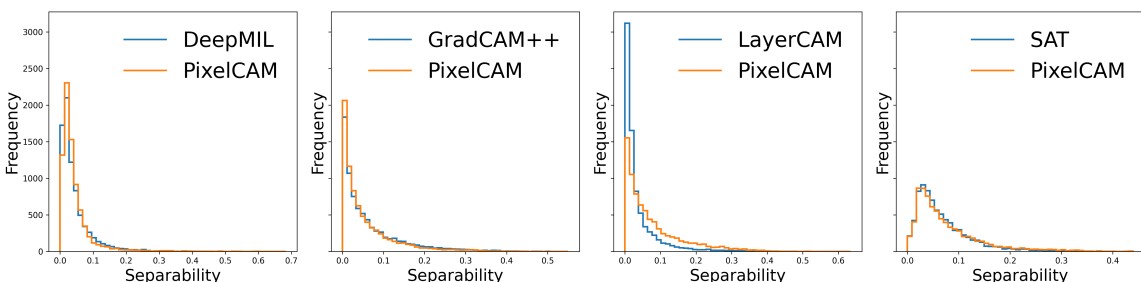

Figure 8: OOD setup: Histogram of class separability index $J$ between FG/BG pixel-features on target test set `CAMELYON16` for the OOD case: `GlaS` → `CAMELYON16` for WSOL baselines with and without `PixelCAM`. The higher $J$ is, the more classes are separated.

## Appendix E. More Localization Performance Analysis

PxAP is the primary metric used to measure localization performance. However, following (Rony et al., 2023), we include other pixel-wise performance measures that is true/false positives/negative rates. As shown previously, our method PixelCAM achieves better results in terms of PxAP performance. This improvement is observed in Tab. 6 with a higher number of true positive/negative rates compared to GradCAM++, LayerCAM, SAT, and NEGEV on the GlaS dataset. For the CAMELYON16 dataset, we observe that PixelCAM increases the true positive rate compared to standard methods while being competitive to NEGEV. However, we note a significant improvement when applying PixelCAM to the SAT method, with a considerable increase in true positives, thereby reducing the over-activation issue.

| | GlaS | | | | CAMELYON16 | | | |
|---|---|---|---|---|---|---|---|---|
| **Bottom-up WSOL** | TP* ↑ | FN* ↓ | TN* ↑ | FP* ↓ | TP* ↑ | FN* ↓ | TN* ↑ | FP* ↓ |
| DeepMIL (Ilse et al., 2018) ICML † | 63.4 | 36.6 | 76.3 | 23.6 | **66.9** | **33.1** | 89.6 | 10.4 |
| DeepMIL w/ NEGEV (Belharbi et al., 2022a) MIDL ⋆ | **79.0** | **21.0** | 75.5 | 24.5 | 64.0 | 36.0 | 92.6 | 7.4 |
| DeepMIL w/ PixelCAM † | 76.4 | 23.6 | **77.4** | **22.6** | 59.0 | 41.0 | **93.3** | **6.7** |
| GradCAM++ (Chattopadhyay et al., 2018) WACV † | 62.0 | 38.0 | **79.9** | **20.1** | 42.1 | 57.8 | 89.4 | 10.6 |
| GradCAM w/ NEGEV (Belharbi et al., 2022a) MIDL ⋆ | 58.4 | 41.6 | 76.0 | 24.0 | **53.5** | **46.5** | **92.2** | **7.8** |
| GradCAM w/ PixelCAM † | **76.9** | **23.1** | 78.8 | 21.2 | 49.2 | 50.8 | 91.4 | 8.6 |
| LayerCAM (Jiang et al., 2021) IEEE TIP † | 62.3 | 37.7 | **72.6** | **27.4** | 29.6 | 70.3 | 86.8 | 13.2 |
| LayerCAM w/ NEGEV (Belharbi et al., 2022a) MIDL ⋆ | 66.3 | 33.7 | 70.7 | 29.3 | 54.4 | 45.6 | **90.9** | **9.1** |
| LayerCAM w/ PixelCAM † | **77.6** | **22.4** | 72.5 | 27.5 | **56.5** | **43.5** | 89.2 | 10.8 |
| U-Net (Ronneberger et al., 2015) MICCAI | 88.9 | 11.1 | 89.8 | 10.2 | 68.0 | 32.0 | 94.5 | 5.5 |

Table 6: Confusion matrix performance over GlaS and CAMELYON16 test set with standard WSOL setup. † refers to model with a single-step approach while ⋆ refers to model from two-step family.

## Appendix F.  Additional Results With Different Backbones

| WSOL models | GlaS | | CAMELYON16 | |
|---|---|---|---|---|
| | PxAP ↑ | CL ↑ | PxAP ↑ | CL ↑ |
| DeepMIL (Ilse et al., 2018) ICML † | 73.3 | **100.0** | 60.3 | **88.0** |
| DeepMIL w/ NEGEV (Belharbi et al., 2022a) MIDL ⋆ | 76.7 | **100.0** | 62.2 | 86.6 |
| DeepMIL w/ PixelCAM † | **78.6** | **100.0** | **69.9** | 85.4 |
| GradCAM++ (Chattopadhyay et al., 2018) WACV † | 72.9 | **100.0** | 27.7 | 87.4 |
| GradCAM++ w/ NEGEV (Belharbi et al., 2022a) MIDL ⋆ | **80.4** | **100.0** | **68.3** | 87.3 |
| GradCAM++ w/ PixelCAM † | 76.9 | **100.0** | 64.9 | 87.7 |
| LayerCAM (Jiang et al., 2021) IEEE TIP † | 73.0 | **100.0** | 24.5 | 88.7 |
| LayerCAM w/ NEGEV (Belharbi et al., 2022a) MIDL ⋆ | **80.7** | **100.0** | **68.7** | 88.1 |
| LayerCAM w/ PixelCAM † | 76.3 | **100.0** | 64.2 | 87.1 |

Table 7:  Localization (PxAP) and classification (CL) accuracy on GlaS and CAMELYON16 test sets using different WSOL methods with VGG16 architecture. † refers to model with a one-step approach while ⋆ refers to model from two-step family.

| WSOL models | GlaS | | CAMELYON16 | |
|---|---|---|---|---|
| | PxAP ↑ | CL ↑ | PxAP ↑ | CL ↑ |
| DeepMIL (Ilse et al., 2018) ICML † | 62.6 | 87.5 | 58.1 | 89.0 |
| DeepMIL w/ NEGEV (Belharbi et al., 2022a) MIDL ⋆ | 66.1 | **91.2** | **76.8** | **89.5** |
| DeepMIL w/ PixelCAM † | **72.0** | 88.8 | 75.7 | 83.2 |
| GradCAM++ (Chattopadhyay et al., 2018) WACV † | 70.0 | 53.8 | 64.0 | 88.4 |
| GradCAM++ w/ NEGEV (Belharbi et al., 2022a) MIDL ⋆ | **81.7** | 93.7 | 74.1 | 86.9 |
| GradCAM++ w/ PixelCAM † | 81.0 | **96.3** | **75.0** | **89.8** |
| LayerCAM (Jiang et al., 2021) IEEE TIP † | 68.3 | 92.5 | 53.7 | **83.8** |
| LayerCAM w/ NEGEV (Belharbi et al., 2022a) MIDL ⋆ | 77.8 | 93.8 | **71.6** | **83.8** |
| LayerCAM w/ PixelCAM † | **78.3** | **100.0** | 65.5 | 82.5 |

Table 8:  Localization (PxAP) and classification (CL) accuracy on GlaS and CAMELYON16 test sets using different WSOL methods with InceptionV3 architecture. † refers to model with a one-step approach while ⋆ refers to model from two-step family.

| WSOL models | GlaS | | CAMELYON16 | |
|---|---|---|---|---|
| | T-stat ↑ | p-value ↓ | T-stat ↑ | p-value ↓ |
| DeepMIL (Ilse et al., 2018) ICML † | 16.8 | $3.4 \times 10^{-16}$ | 9.3 | $5.1 \times 10^{-10}$ |
| GradCAM++ (Chattopadhyay et al., 2018) WACV † | 11.2 | $7.9 \times 10^{-12}$ | 139.1 | $2.1 \times 10^{-41}$ |
| LayerCAM (Jiang et al., 2021) IEEE TIP † | 6.6 | $3.9 \times 10^{-7}$ | 180.1 | $1.9 \times 10^{-44}$ |

Table 9:  T-test statistics between baselines (DeepMIL, GradCAM++, LayerCAM) and our method (PixelCAM) for localization performance with VGG16 backbone.

| WSOL models | GlaS | | CAMELYON16 | |
|---|---|---|---|---|
| | T-stat ↑ | p-value ↓ | T-stat ↑ | p-value ↓ |
| DeepMIL (Ilse et al., 2018) ICML † | 17.8 | $8.2 \times 10^{-17}$ | 18.0 | $6.5 \times 10^{-17}$ |
| GradCAM++ (Chattopadhyay et al., 2018) WACV † | 19.3 | $1.0 \times 10^{-17}$ | 10.9 | $1.5 \times 10^{-11}$ |
| LayerCAM (Jiang et al., 2021) IEEE TIP † | 13.7 | $5.8 \times 10^{-14}$ | 11.0 | $1.1 \times 10^{-11}$ |

Table 10:  T-test statistics between baselines (DeepMIL, GradCAM++, LayerCAM) and our method (PixelCAM) for localization performance with InceptionV3 backbone.

## Appendix G. Computational Complexity

The computation overhead of the pixel classifier in `PixelCAM` is minimal in our context. It is suitable for analyzing whole slide images (WSIs) that contain millions of pixels (Rony et al., 2023). From a memory point of view, the impact is negligible since the number of supplementary parameters depends essentially on feature size. For instance, in the ResNet50-based architecture, the feature size is equal to 2048, and the number of classes is equal to two (FG and BG). The pixel classifier adds only 4,098 parameters to the CNN-based architecture, which already contains more than 23.5M parameters. The same conclusion applies to the transformer-based architecture ($> 5.5$M parameters), where we add 386 parameters only.

For the inference computational cost, our pixel classifier is particularly advantageous, especially compared to WSOL gradient-based models for WSOL (see Tab.11). Therefore, our model incurs negligible computation overhead, allowing for fast training and inference. The time efficiency, combined with the robustness of PixelCAM, makes it a key advantage for deployment in practical scenarios where WSI image sizes can be extremely large.

| WSOL Models | Inference time ↓ | No. parameters |
|---|---|---|
| DeepMIL | **9.1**ms | 24,036,932 |
| DeepMIL w/ NEGEV | 12.3ms | 33,050,150 |
| DeepMIL w/ PixelCAM | 9.2ms | 24,041,030 |
| GradCAM ++ | 40.9ms | 23,514,179 |
| GradCAM ++ w/ NEGEV | 10.3ms | 32,527,397 |
| GradCAM ++ w/ PixelCAM | **8.8**ms | 23,518,277 |
| LayerCAM | 37.9ms | 23,514,179 |
| LayerCAM w/ NEGEV | 10.4ms | 32,527,397 |
| LayerCAM w/ PixelCAM | **8.4**ms | 23,518,277 |
| SAT | **10.5**ms | 5,528,267 |
| SAT w/ PixelCAM | 11.5ms | 5,528,651 |
| U-Net | **10.1**ms | 32,521,250 |

Table 11: Inference time required to produce CAMs using different WSOL methods with a ResNet50 architecture for CNN-based models and DeiT-Tiny for Transformer-based model. The time needed to build a full-size CAM is estimated using an NVIDIA RTX A6000 GPU for one random RGB image of size $224 \times 224$.

## Appendix H. Additional Results for Classification (CL) and Localization (PxAP) Accuracy on CAMELYON16 Test Set with Different Stainings.

In this section, we extend the experiment initially conducted on the `GlaS` dataset to the `CAMELYON16` dataset. We modify the stainings in the `CAMELYON16` test set to evaluate the robustness of `PixelCAM` compare to the baseline method (LayerCAM). `PixelCAM` consistently outperforms the baseline in terms of robustness on both classification and localization tasks.

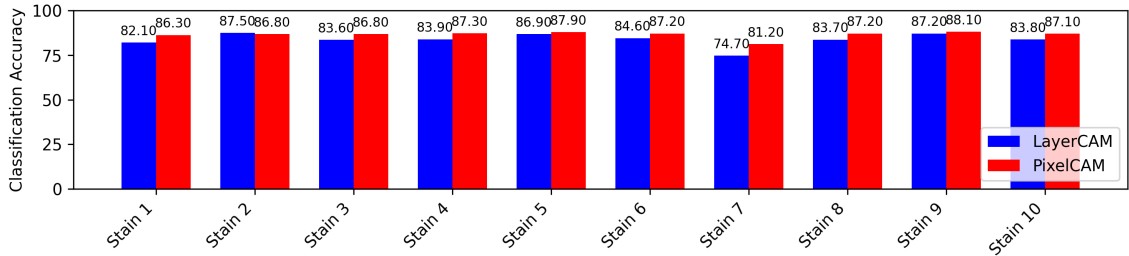

Figure 9: Classification (`CL`) accuracy on `CAMELYON16` test sets with LayerCAM and `PixelCAM` with different stainings.

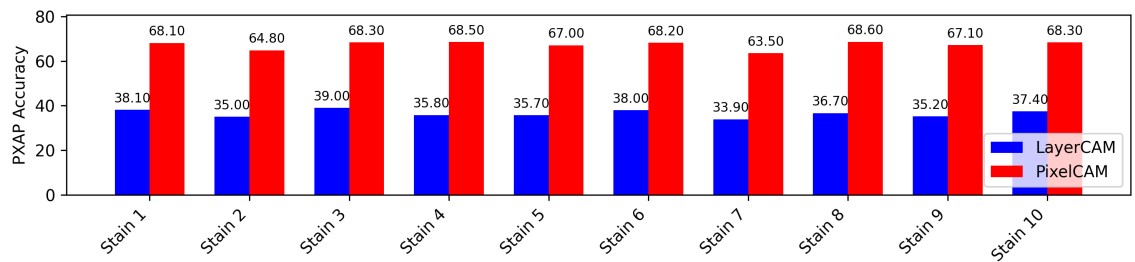

Figure 10: Localization (`PxAP`) accuracy on `CAMELYON16` test sets with LayerCAM and `PixelCAM` with different stainings.

## Appendix I. Limitations and Future Work

WSOL methods are known to underperform when applied to new datasets due to domain shift (Guichemerre et al., 2024). Although our model is impacted by this issue, it demonstrates improved performance in terms of `PxAP` (localization) and `CL` (classification). However, it may still face difficulties in maintaining the same level of performance as on the source dataset, particularly in the presence of extreme shifts (e.g., from `GlaS` to `CAMELYON16`). To mitigate these limitations, several domain adaptation strategies could be explored. Many existing approaches rely on feature alignment techniques, such as contrastive learning or distribution alignment, to reduce domain shift. Our new pixel classifier can serve exactly as the image classifier by adapting such techniques at the pixel level. Additionally, in domain adaptation context, most of the techniques use clustering techniques to refine pseudo labels. Since our model produces more discriminative features (as shown in Tables 4 and 5 ), `PixelCAM` can improve clustering effectiveness leading to more reliable pixel pseudo-label adaptation approaches.

## Appendix J. Ablations

We provide in this section several ablations of our method `PixelCAM`.

### J.1. Impact of `PixelCAM` Depth

All the reported results of our method are obtained with a linear classifier that classifies pixel-embeddings into FG/BG classes. In this section, we further explore the impact of using a multi-layer classifier composed of three 1×1 convolutional layers, which act as fully connected layers by reducing the dimensionality of the previous layer's output by a factor of 2.

| | GlaS | | CAMELYON16 | |
|---|---|---|---|---|
| **WSOL models** | PxAP ↑ | CL ↑ | PxAP ↑ | CL ↑ |
| DeepMIL (Ilse et al., 2018) ICML | 79.9 | **100.0** | 71.3 | 85.0 |
| DeepMIL w/ linear `PixelCAM` | 85.5 | **100.0** | **75.7** | 88.2 |
| DeepMIL w/ multi-layer `PixelCAM` | **86.3** | 95.0 | 74.4 | **88.6** |
| GradCAM++ (Chattopadhyay et al., 2018) WACV | 77.9 | **100.0** | 49.1 | 63.4 |
| GradCAM++ w/ linear `PixelCAM` | **86.6** | **100.0** | 63.4 | **88.7** |
| GradCAM++ w/ multi-layer `PixelCAM` | 86.5 | 95.0 | **64.0** | 85.8 |

Table 12: Localization (`PxAP`) and classification (`CL`) accuracy on `GlaS` and `CAMELYON16` test set with linear and multi-layer `PixelCAM` for Standard WSOL setup.

As shown in Tab. 12, adding hidden layers to the pixel classifier is not necessarily an advantage for the localization task. Indeed, models with multiple layers perform worse on `GlaS` and `CAMELYON16` for the WSOL baseline methods GradCAM++ (Chattopadhyay et al., 2018) and DeepMIL (Ilse et al., 2018), respectively. In the case of OOD data, Tab.13 shows that large performance degradation can observed when using multi-layer classifier. Therefore, as a general rule, we recommend using simply a linear pixel-classifier.

| | CAMELYON16 → GlaS | | GlaS → CAMELYON16 | |
|---|---|---|---|---|
| **WSOL models** | PxAP ↑ | CL ↑ | PxAP ↑ | CL ↑ |
| DeepMIL (Ilse et al., 2018) ICML | 64.5 | 81.2 | 29.0 | **55.2** |
| DeepMIL w/ linear `PixelCAM` | **69.1** | **83.8** | **30.2** | 52.5 |
| DeepMIL w/ multi-layer `PixelCAM` | 67.4 | 80.0 | 25.0 | 51.6 |
| GradCAM++ (Chattopadhyay et al., 2018) WACV | 52.9 | 53.7 | **39.1** | 52.4 |
| GradCAM++ w/ linear `PixelCAM` | **56.2** | **71.2** | 36.9 | **63.3** |
| GradCAM++ w/ multi-layer `PixelCAM` | 55.8 | **71.2** | 33.8 | 50.6 |

Table 13: Localization (`PxAP`) and classification (`CL`) accuracy on `GlaS` and `CAMELYON16` test set with linear and multi-layer `PixelCAM` for OOD setup.

**J.2. Impact of Model Selection of CAM Pre-trained Model**

Our method leverages pseudo-labels extracted from a pretrained WSOL CAM-based model. However, due to the asynchronous convergence of classification and localization tasks (Rony et al., 2023), the criterion used for model selection is expected to have an impact on CAM quality, and therefore, pseudo-labels accuracy. Typically, models are selected based either on their classification performance on validation set by taking the best classifier (B-CL), or their localization performance and considering the best localizer (B-LOC). In Tab.14, we present the impact of such choice on our method PixelCAM.

| | GlaS | | CAMELYON16 | |
| :--- | :---: | :---: | :---: | :---: |
| **WSOL models** | PxAP ↑ | CL ↑ | PxAP ↑ | CL ↑ |
| LayerCAM (Jiang et al., 2021) IEEE TIP | 75.1 | **100.0** | 33.2 | 84.8 |
| LayerCAM w/ PixelCAM (B-LOC CAM) | **83.6** | **100.0** | 66.2 | 89.1 |
| LayerCAM w/ PixelCAM (B-CL CAM) | 80.6 | 98.8 | **67.5** | **89.4** |
| SAT (Wu et al., 2023) ICCV | 65.9 | 98.8 | 32.8 | 83.2 |
| SAT w/ PixelCAM (B-LOC CAM) | **79.1** | **100.0** | **51.2** | **87.2** |
| SAT w/ PixelCAM (B-CL CAM) | 75.5 | **100.0** | 35.6 | 86.4 |

Table 14: Impact of model selection of CAM-based model to build pseudo-labels: Comparison of PixelCAM on GlaS and CAMELYON16 datasets on test set by using CAMs from B-LOC and B-CL models respectively. The performance is compared to the standard WSOL setup.

Initially, the CAMs used for pixel selection were generated from the B-LOC model. We compared the performance compared to using B-CL's CAMs. Table 14 shows the results of various experiments with different WSOL baseline methods. We noticed that PixelCAM improves localization performance regardless of whether B-LOC or B-CL is used. However, for techniques that use gradients, such as GradCAM++ (Chattopadhyay et al., 2018) and LayerCAM (Jiang et al., 2021), CAMs generated with B-CL provide better performance. This can be explained by the nature of the CAMs produced by B-LOC for these methods. Specifically, B-LOC generates CAMs with better localization performance due to higher true positive rates but also introduces more false positives. This negatively impacts training, as incorrect labels are incorporated, potentially reducing overall performance.

We also observe a significant difference for SAT method (Wu et al., 2023). This is mainly explained by the fact that the CAMs generated by the B-CL model produce a high number of false positives, similar to those generated by B-LOC. However, the B-LOC model generates a higher true positive rate than B-CL, which is crucial for the strong performance of PixelCAM.

**J.3. Impact of Pixel Sampling Technique**

Our method PixelCAM uses pseudo-annotation for the pixel classifier's training. In particular, we consider random sampling of pixel locations to generate pseudo-labels as it has shown to yield better performance than fitting static regions (Belharbi et al., 2022a). This avoids

overfitting regions and promote exploring potential ROIs. In this section, we investigate the impact of various sampling techniques to identify pixels associated to FG and BG. We consider two different sampling approaches namely *Threshold-based* (Belharbi et al., 2022c) and *Probability-based (PB)* (Belharbi et al., 2022a). The *Threshold-based (TH)* approach automatically thresholds the CAM. Then, it considers all the pixels with activation above the threshold as foreground and valid for FG sampling. This is delineated by a mask. FG pixels are sampled uniformly within that mask, while BG pixels are sampled from outside the mask. The *Probability-based* approach samples FG pixels proportionally to the probability values obtained from the CAM using a multinomial distribution. However, the BG pixels are samples from (1 - CAM) activations so regions with low activations will have higher sampling chance.

| | GlaS | | CAMELYON16 | |
|---|---|---|---|---|
| **WSOL models** | PxAP ↑ | CL ↑ | PxAP ↑ | CL ↑ |
| GradCAM++ (Chattopadhyay et al., 2018) WACV | 76.8 | **100.0** | 49.1 | 63.4 |
| GradCAM++ w/ TH PixelCAM | 85.4 | **100.0** | 54.2 | **89.9** |
| GradCAM++ w/ PB PixelCAM | **86.6** | **100.0** | **64.1** | 85.1 |
| SAT (Wu et al., 2023) ICCV | 65.9 | 98.8 | 32.8 | 83.2 |
| SAT w/ TH PixelCAM | 71.5 | 92.5 | 50.9 | 81.0 |
| SAT w/ PB PixelCAM | **79.1** | **100.0** | **51.2** | **87.2** |

Table 15: Impact of pixel sampling technique on PixelCAM performance: Threshold-based (TH) vs. probability-based (PB), over standard WSOL setup.

In Tab. 15, we observe that pixel sampling technique has a significant impact on the localization performance of PixelCAM. For GlaS dataset, using GradCAM++ CAMs has a small impact, but for CAMs generated from a transformer architecture as SAT the impact is significant with a difference of 7.6%. Similarly, on a more challenging dataset as CAMELYON16, using a probabilistic approach for sampling pixels can lead to an improvement of 9.9%. This can be explained by the fact that sampling without relying on a threshold favors relevant pixels and most likely to have the correct pseudo-label. On the other hand, threshold-based method fixes a region for sampling with high likelihood of covering wrong regions. Learning with incorrect labels leads to poor models, and, therefore poor performance.

### J.4. Impact of the Number of Sampled Pixels $n$

We analyze the impact of the number of pixels selected as pseudo-labels to train PixelCAM. To avoid unbalanced pixel classification, we sample the same number $n$ of pixels as FG and BG. We consider the following cases $n \in \{1, 5, 10, 20\}$. As observed in Tab. 16, increasing $n$ can affect both localization and classification with different degree, and depending on the dataset. In terms of localization, both datasets can slightly be affected. However, CAMELYON16 is largely affected in term of classification as performance can vary between 85.7% and 90.1%.

| WSOL models | GlaS | | CAMELYON16 | |
|---|---|---|---|---|
| | PxAP ↑ | CL ↑ | PxAP ↑ | CL ↑ |
| GradCAM++ (Chattopadhyay et al., 2018) WACV | 76.8 | **100.0** | 49.1 | 63.4 |
| GradCAM++ w/ `PixelCAM`: | | | | |
| $n = 1$ | **86.7** | **100.0** | 63.5 | **90.1** |
| $n = 5$ | 86.6 | **100.0** | 64.1 | 85.1 |
| $n = 10$ | 86.0 | **100.0** | 63.2 | 88.8 |
| $n = 20$ | 86.0 | **100.0** | **64.7** | 88.7 |

Table 16: Impact of the number of sampled pixel as pseudo-labels $n$ on `PixelCAM` performance. We measure the `PxAP` and `CL` performance over the test set for `GlaS` and `CAMELYON16` for standard WSOL setup.

### J.5. Impact of $\lambda$

Table 17 shows the impact of $\lambda$ on the performance of our method `PixelCAM`. It achieves better results when using a higher value of $\lambda$ on `GlaS`. As $\lambda$ decreases, the `PxAP` performance also declines. On the `CAMELYON16` dataset, we note that using a low $\lambda$ value (0.001) significantly impacts localization performance. Therefore, we recommend using $\lambda$ values in the range of 0.1 to 1.0 for better performance.

| $\lambda$ | GlaS | | CAMELYON16 | |
|---|---|---|---|---|
| | PxAP ↑ | CL ↑ | PxAP ↑ | CL ↑ |
| 1 | **83.6** | **100.0** | 66.2 | 89.1 |
| 0.5 | 83.1 | **100.0** | 66.6 | **89.7** |
| 0.1 | 82.2 | **100.0** | **67.4** | 88.3 |
| 0.01 | 79.3 | 96.3 | 66.7 | 89.1 |
| 0.001 | 76.1 | 98.8 | 61.6 | 89.2 |

Table 17: Impact of hyper-parameter $\lambda$ over `PixelCAM` in terms of `PxAP` and `CL` performance over test set. `PixelCAM` uses LayerCAM (Jiang et al., 2021) for pseudo-labels.

### J.6. Impact of using a pretrained WSOL model on same and external dataset for pseudo-labeling

As mentionned, PixelCAM use a WSOL CAM-based model to obtain pseudo label for foreground and background. In our paper, we consider training a standard classic WSOL CAM-based (DeepMIL, GradCAM++, LayerCAM, SAT) trained on the same dataset to obtain a robust model for the pseudo labeling to avoid the number of false positive. Considering a WSOL CAM-based model trained on an external dataset can considered to avoid computation time during training as it doesn't require a pre step training but tends to perform poorly on a unseen dataset which will cause a high number of false positive and negative leading to a poor pseudo labeling. To support this claim we trained LayerCAM

on `GlaS` to obtain pseudo label on `CAMELYON16` and also trained the model on `CAMELYON16` to obtain pseudo labels on `GlaS`.

| | GlaS | | | | CAMELYON16 | | | |
| --- | --- | --- | --- | --- | --- | --- | --- | --- |
| | Same | | External | | Same | | External | |
| | PxAP ↑ | CL ↑ | PxAP ↑ | CL ↑ | PxAP ↑ | CL ↑ | PxAP ↑ | CL ↑ |
| LayerCAM | 75.1 | **100.0** | 58.1 | 83.8 | 33.2 | 84.8 | 22.7 | 50.4 |
| LayerCAM w/ PixelCAM | **83.6** | **100.0** | **68.1** | **92.5** | **66.2** | **89.1** | **59.2** | **81.7** |

Table 18: Localization (`PxAP`) and classification (`CL`) accuracy on `GlaS` and `CAMELYON16` test sets with VGG16 and ResNet50 backbones.

As we can observe (Tab.18), the performance of using a WSOL CAM-based method on a external avoid computation cost on the training part but significantly impact the training of PixelCAM and make it less suitable compare to a standard WSOL CAM-based method on GlaS dataset.

## Appendix K. More Visualization

### K.1. Visualization Results on Standard WSOL setup

In this section, we present visual results of `PixelCAM` compared to WSOL baseline methods. In a standard WSOL setup, we observe that `PixelCAM` enhances the ROIs detected by WSOL baseline methods on the `GlaS` dataset (Fig: 11 and 12). Regarding `CAMELYON16`, a challenging dataset, `PixelCAM` improves localization for cancer images by extending ROIs and reducing incorrect predictions from WSOL baseline methods as observed in (Fig: 13).

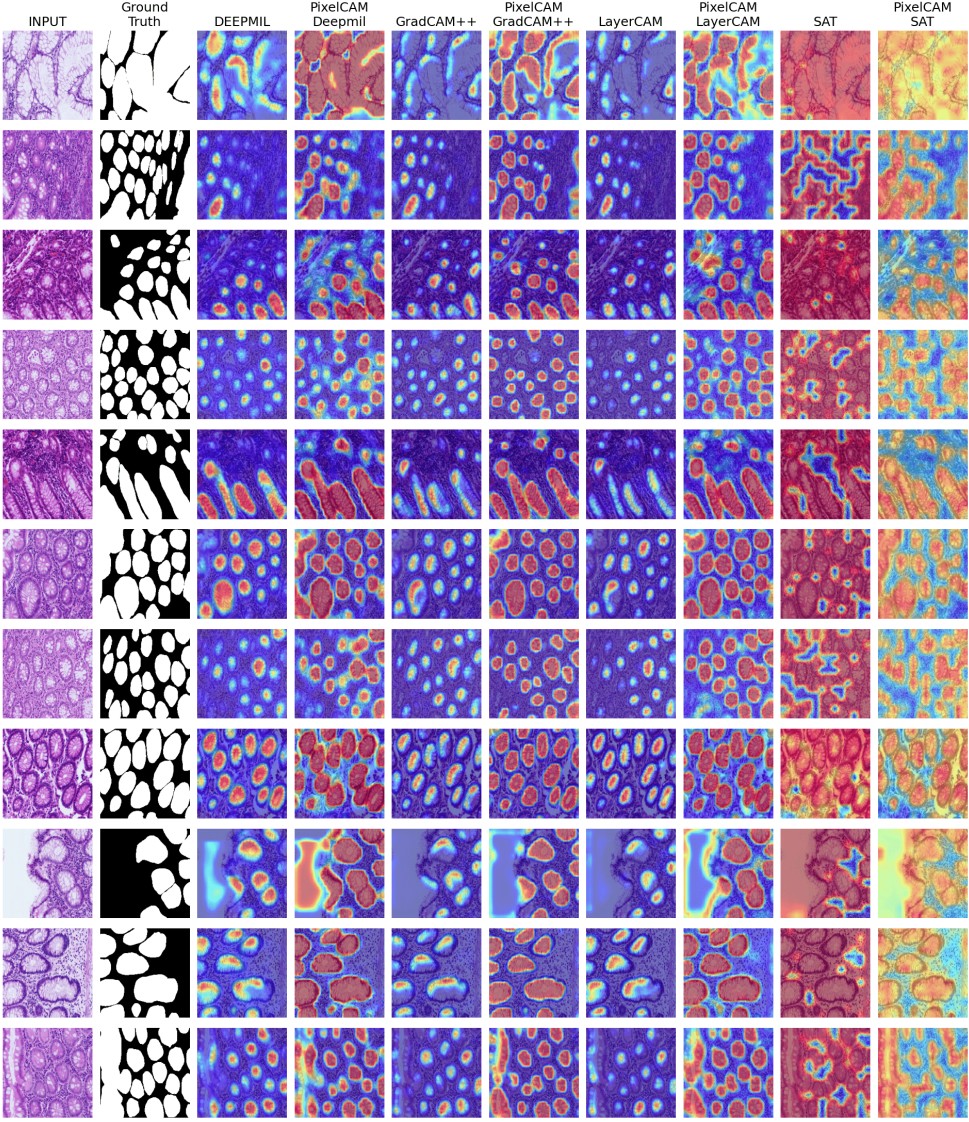

Figure 11: Standard WSOL setup. First column: **Normal** images from `GlaS`. Second column: Ground truth. Next columns: We display the visual CAM results for WSOL baseline without and with `PixelCAM`, respectively.

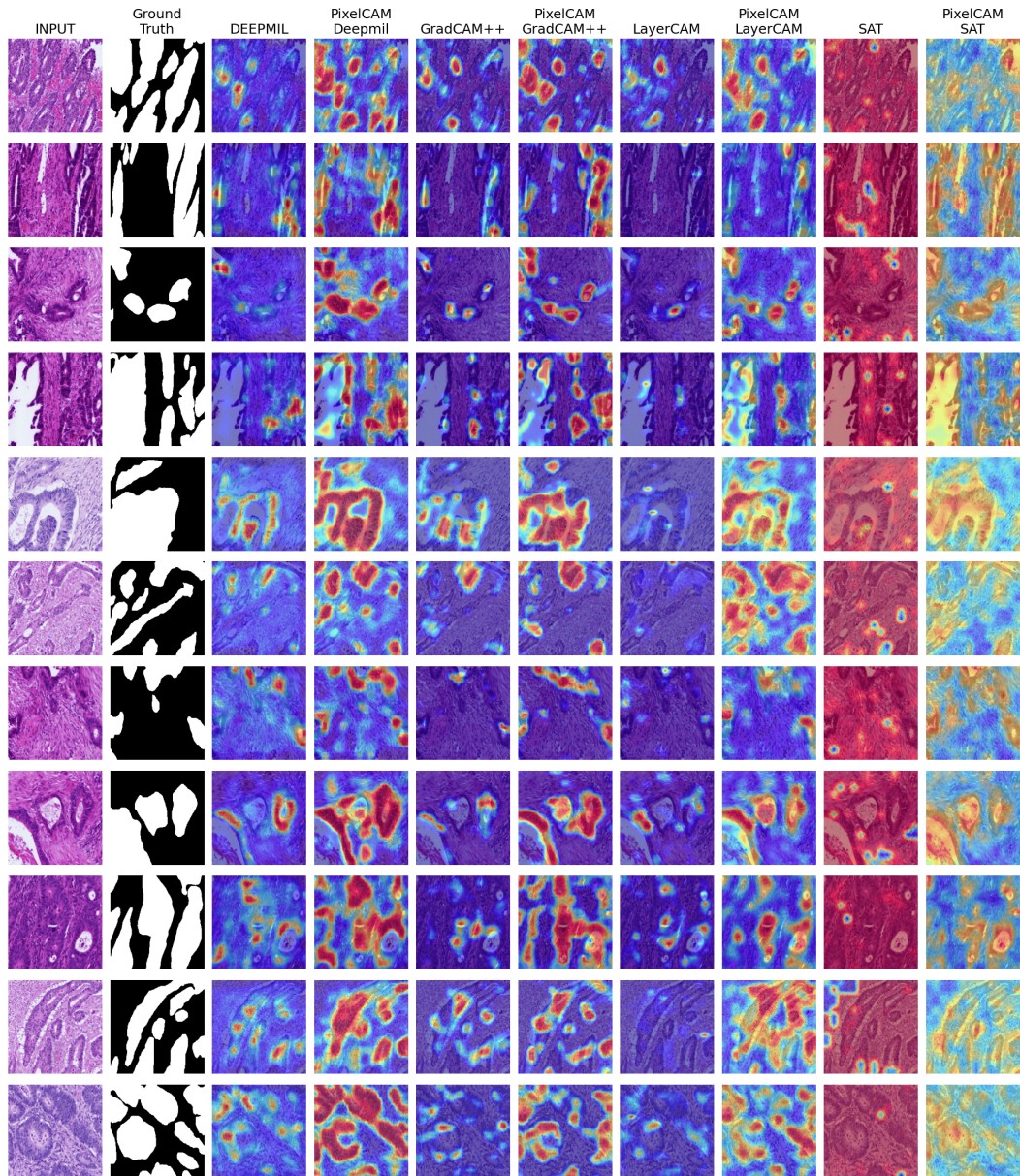

Figure 12: Standard WSOL setup. First column: **Cancerous** images from `GlaS`. Second column: Ground truth. Next columns: We display the visual CAM results for WSOL baseline without and with `PixelCAM`, respectively.

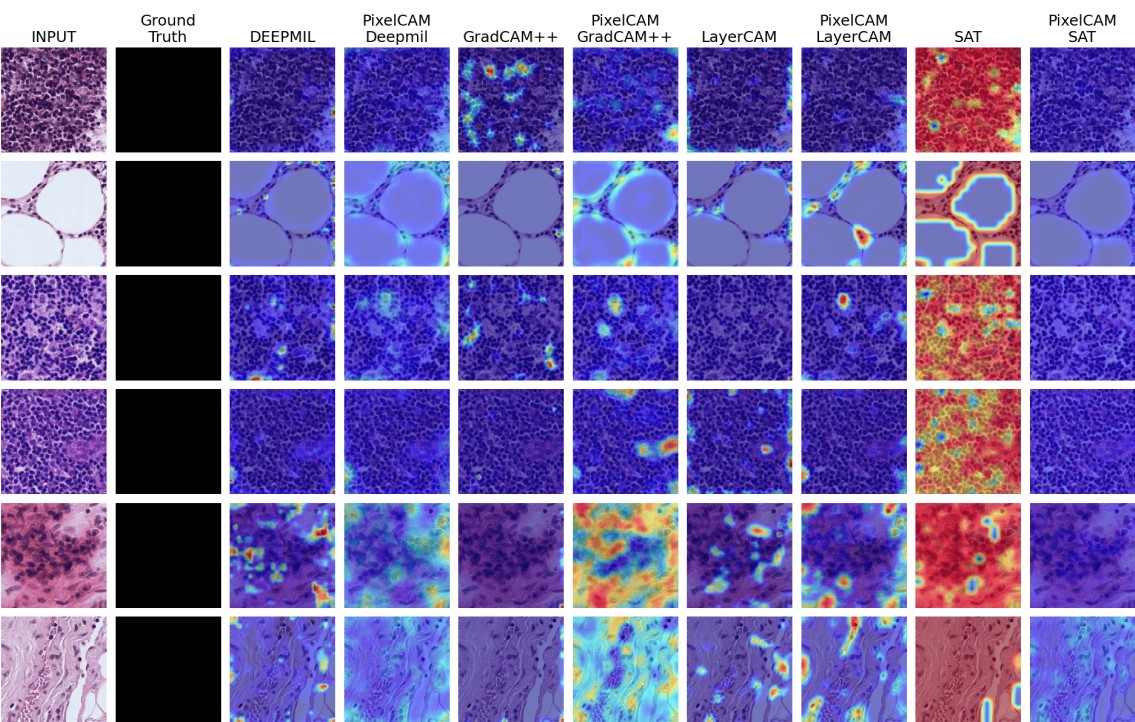

Figure 13: Standard WSOL setup. First column: **Normal** images from `CAMELYON16`. Second column: Ground truth. Next columns: We display the visual CAM results for WSOL baseline without and with `PixelCAM`, respectively.

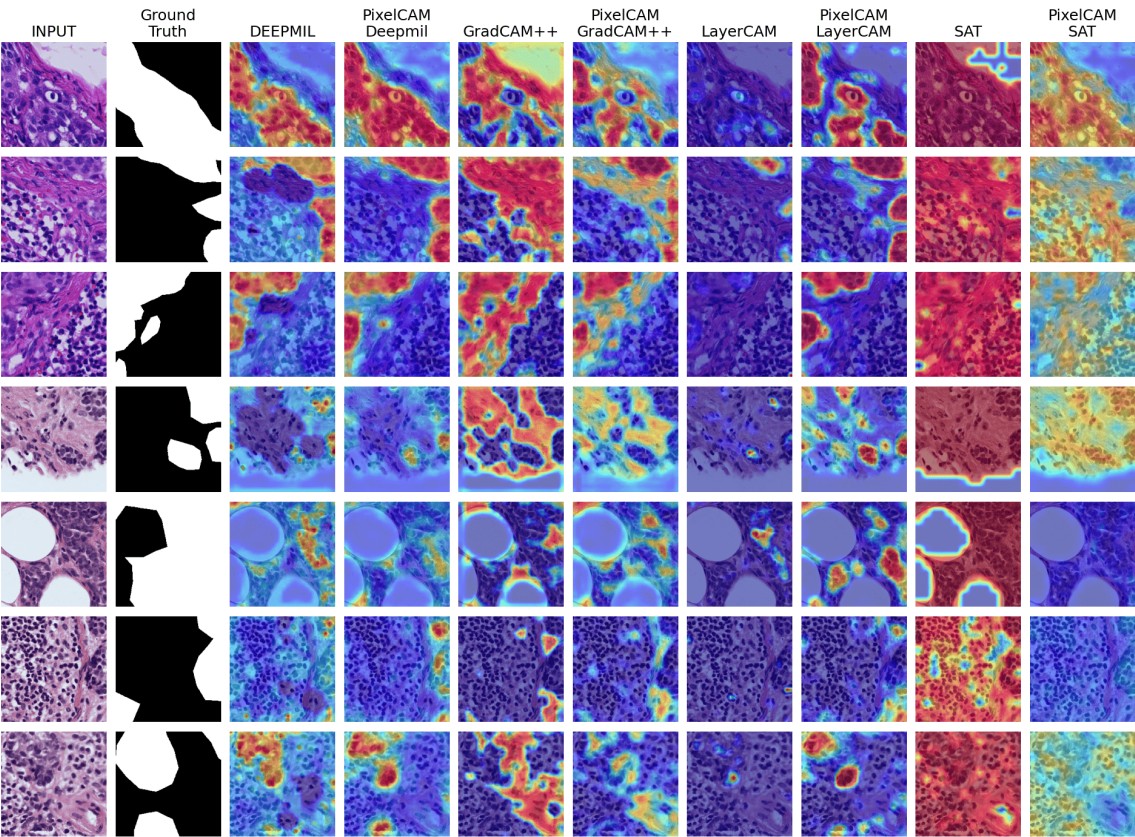

Figure 14: Standard WSOL setup. First column: **Cancerous** images from `CAMELYON16`. Second column: Ground truth. Next columns: We display the visual CAM results for WSOL baseline without and with `PixelCAM`, respectively.

## K.2. Visualization Results on Target set with OOD setup

We provide visual results on target set in the case of OOD for both scenarios: `CAMELYON16 → GlaS` and `GlaS → CAMELYON16`. As we observe, `PixelCAM` can predict correctly in average cancer ROIs as illustrated in Fig: 16 and 18 but struggle with normal images as the WSOL baseline methods (Fig: 15 and 17).

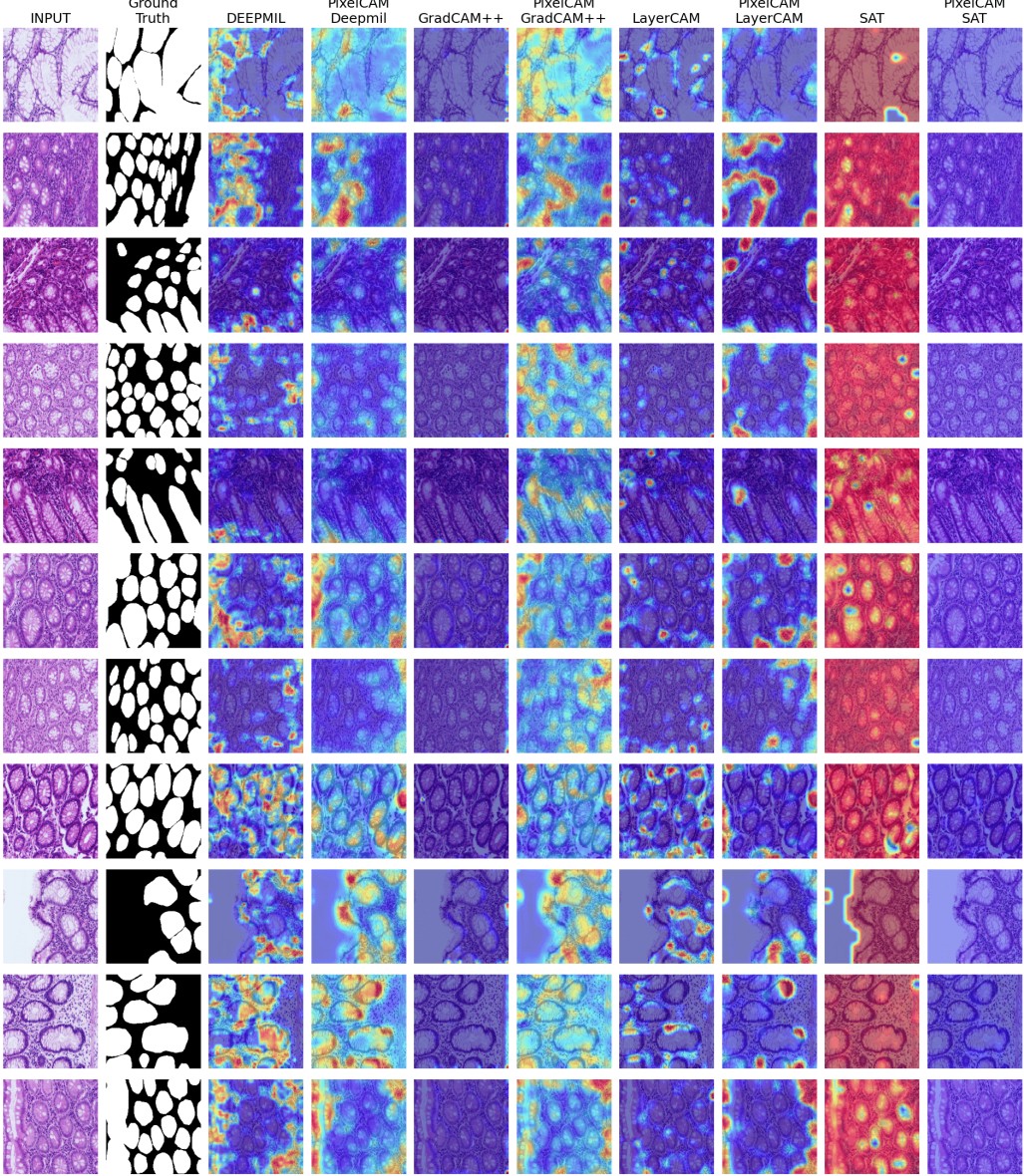

Figure 15: OOD setup: `CAMELYON16 → GlaS`. First column: **Normal** images from `GlaS`. Second column: Ground truth. Next columns: We display the visual CAM results for WSOL baseline without and with `PixelCAM`, respectively.

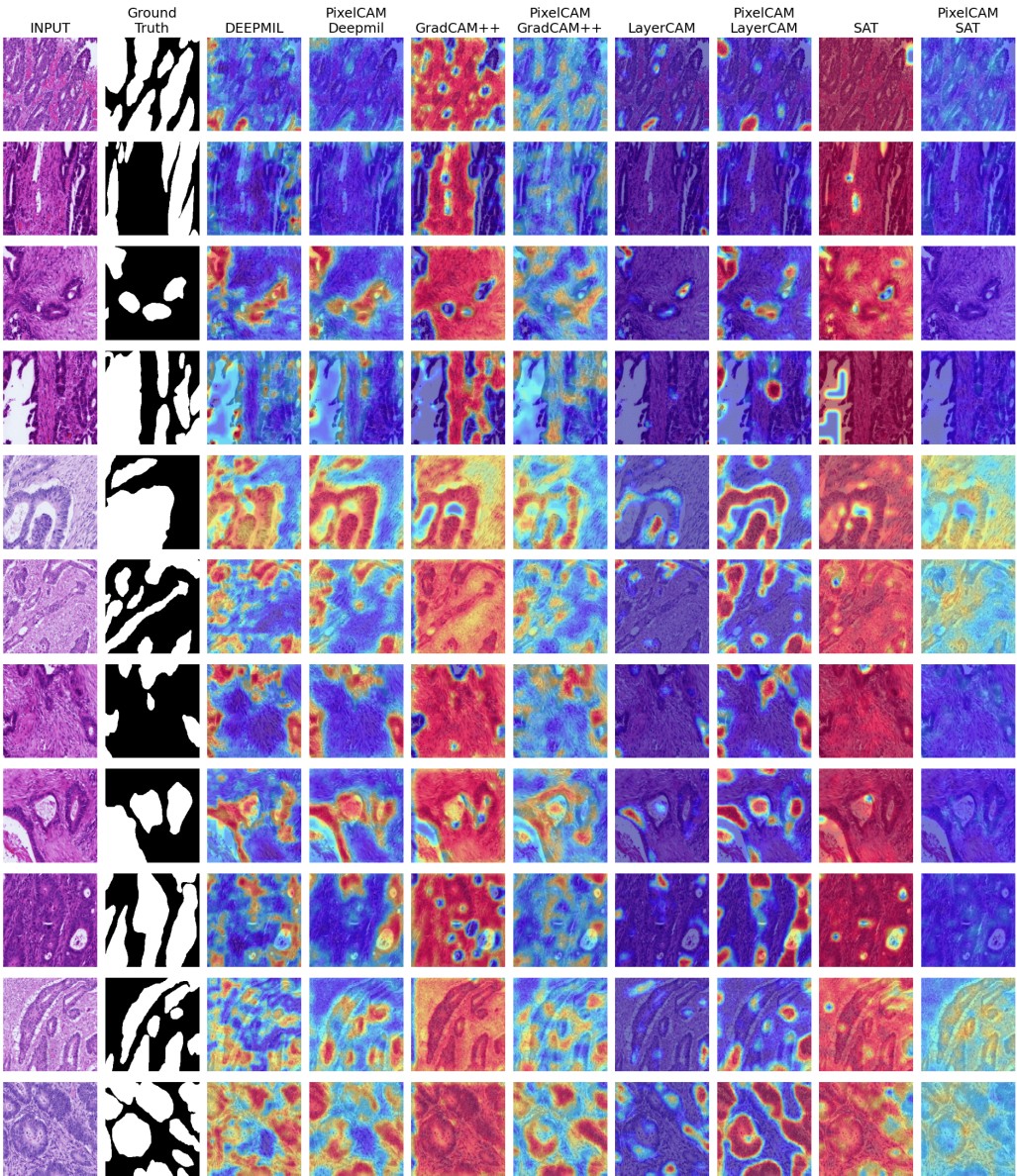

Figure 16: OOD setup: `GlaS` → `CAMELYON16`. First column: **Cancerous** images from `GlaS`. Second column: Ground truth. Next columns: We display the visual CAM results for WSOL baseline without and with `PixelCAM`, respectively.

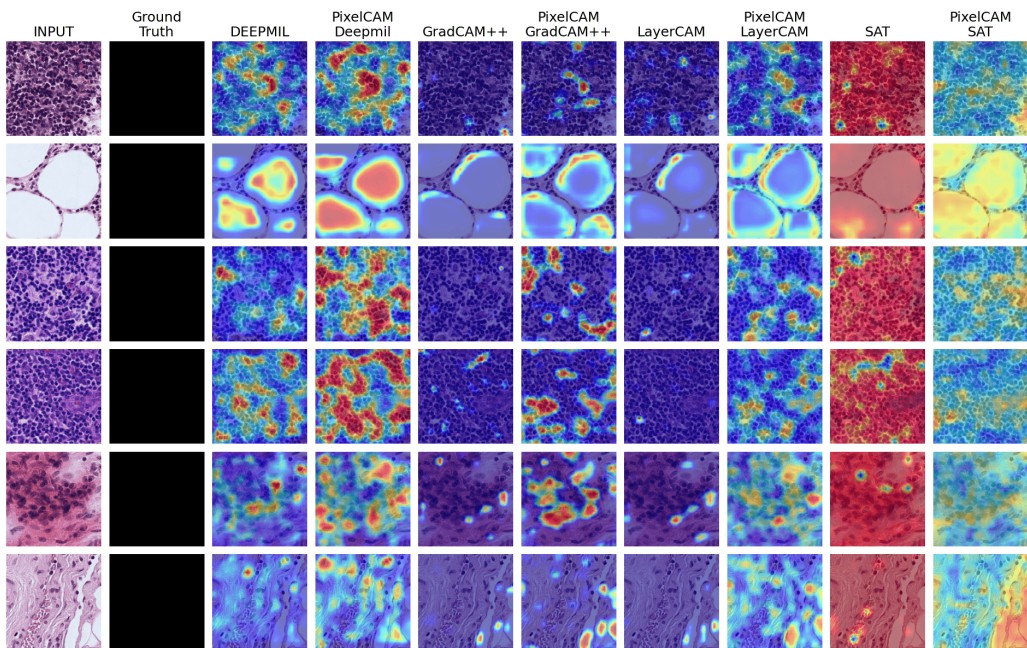

Figure 17: OOD setup: GlaS → CAMELYON16. First column: **Normal** images from CAMELYON16. Second column: Ground truth. Next columns: We display the visual CAM results for WSOL baseline without and with PixelCAM, respectively.

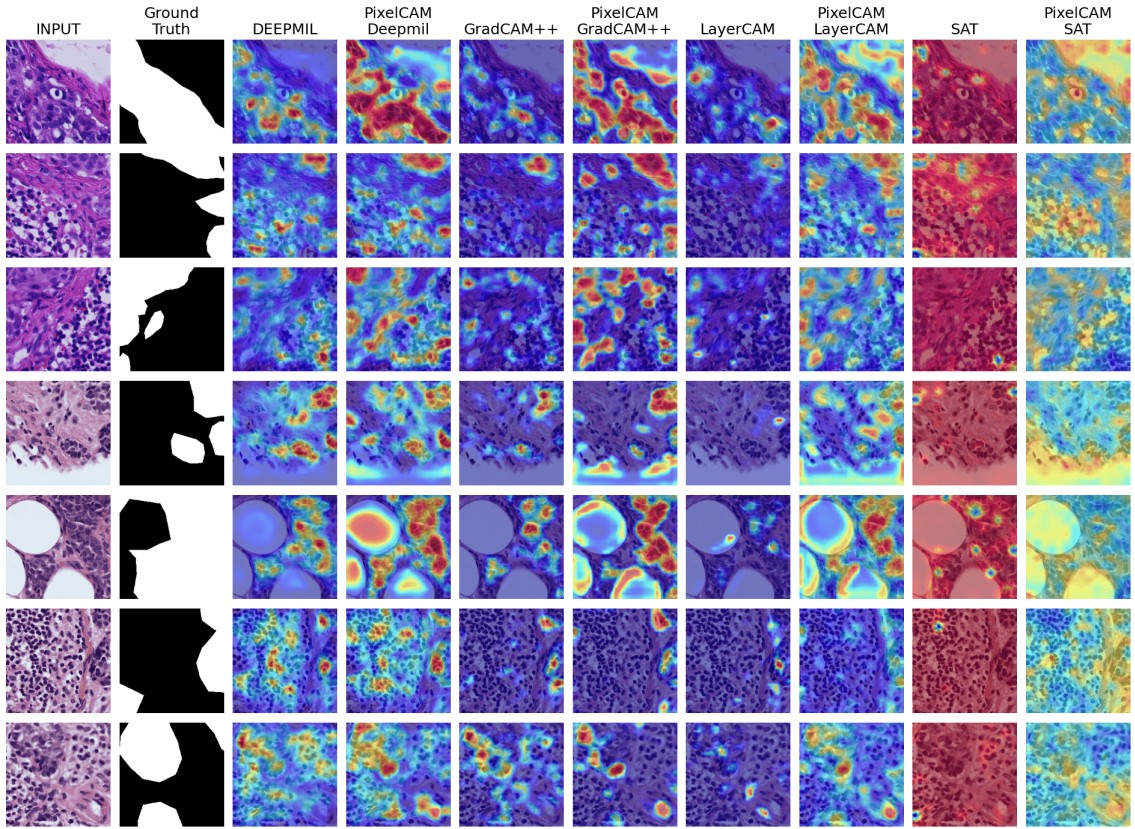

Figure 18: OOD setup: `GlaS` → `CAMELYON16`. First column: **Cancerous** images from `CAMELYON16`. Second column: Ground truth. Next columns: We display the visual CAM results for WSOL baseline without and with `PixelCAM`, respectively.

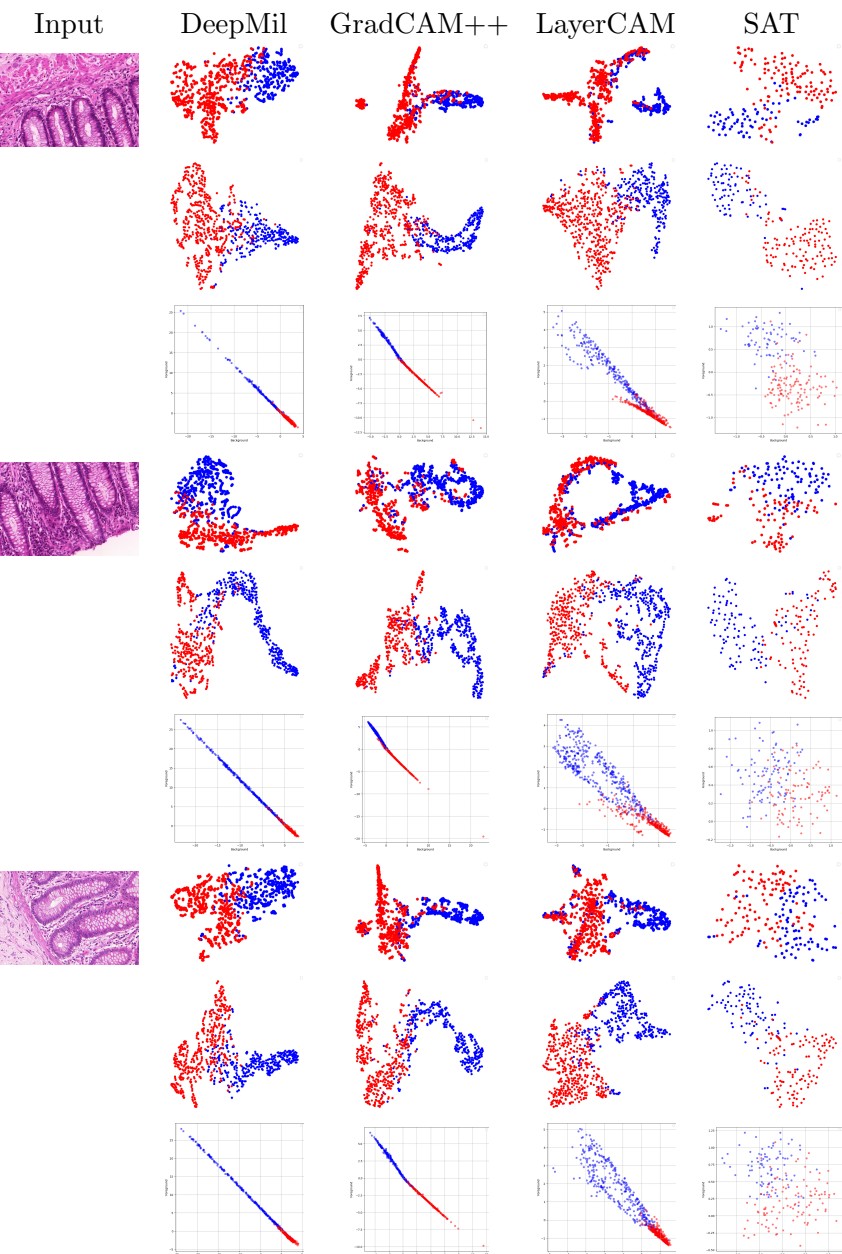

Figure 19: Standard WSOL setup: For the three **normal** images from `GlaS`, we display the t-SNE projection of foreground and background pixel-features of WSOL baseline methods without (1st row) and with (2nd row) `PixelCAM`. The 3rd row presents the logits of the pixel classifier of `PixelCAM`.

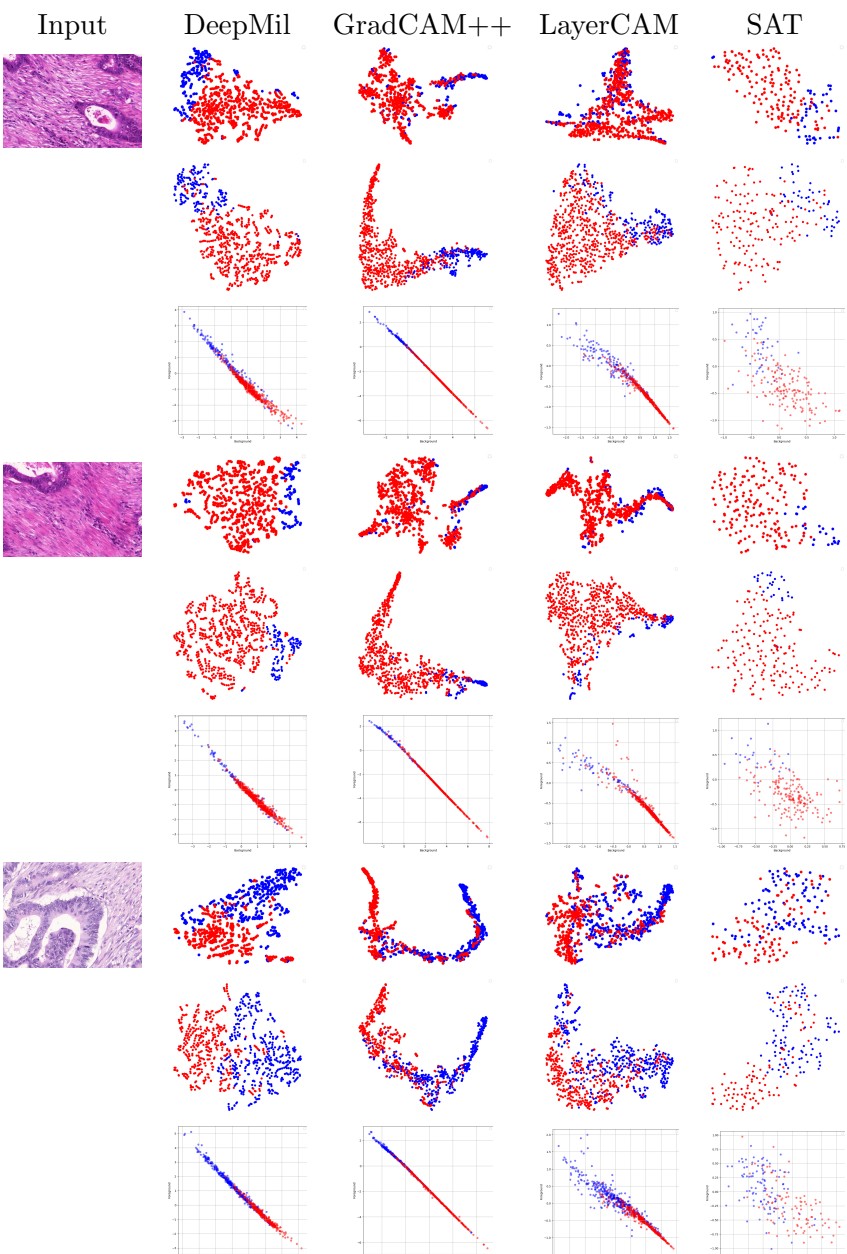

Figure 20: Standard WSOL setup: For the three **cancerous** images from `GlaS`, we display the t-SNE projection of foreground and background pixel-features of WSOL baseline methods without (1st row) and with (2nd row) `PixelCAM`. The 3rd row presents the logits of the pixel classifier of `PixelCAM`.

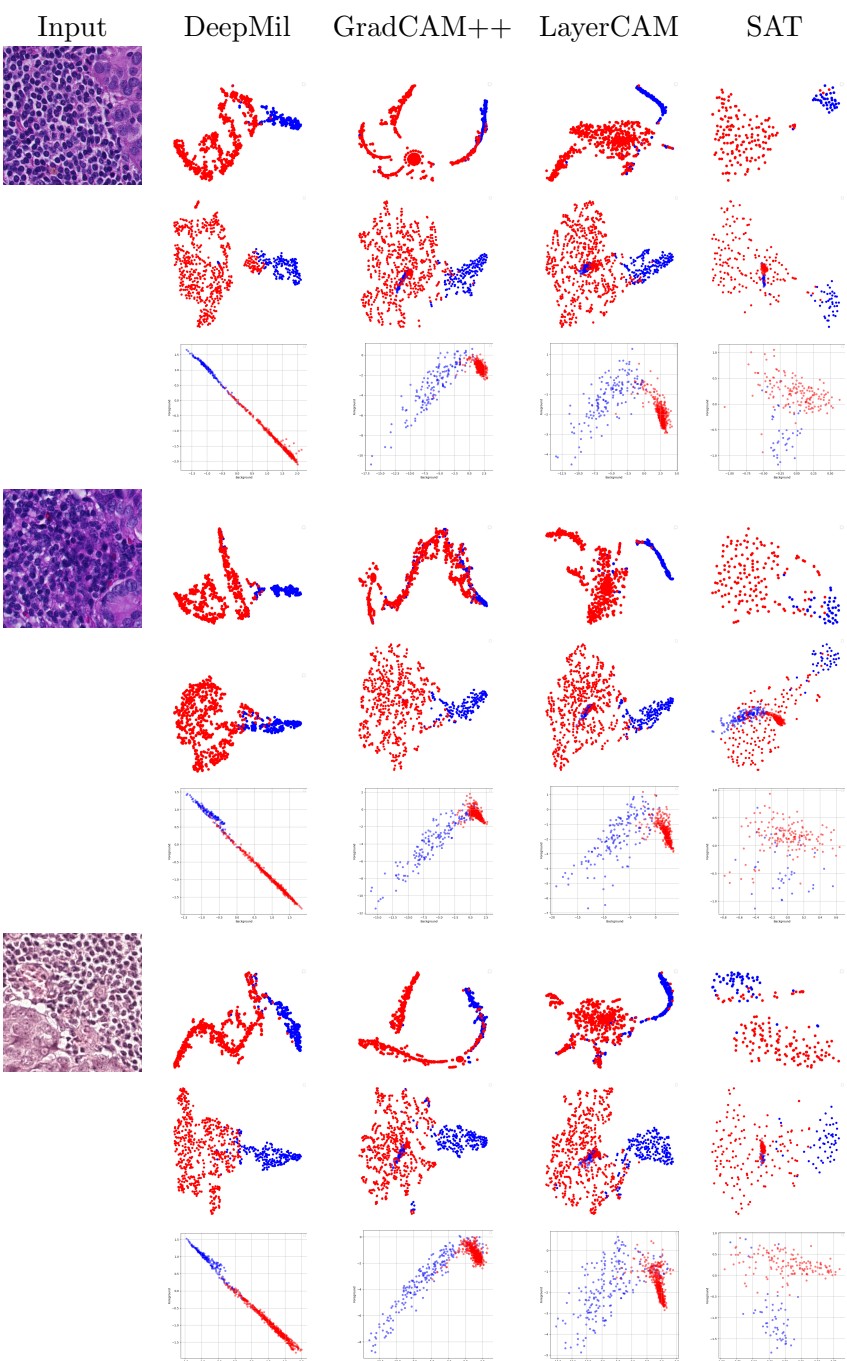

Figure 21: Standard WSOL setup: For the three **cancerous** images from `CAMELYON16`, we display the t-SNE projection of foreground and background pixel-features of WSOL baseline methods without (1st row) and with (2nd row) `PixelCAM`. The 3rd row presents the logits of the pixel classifier of `PixelCAM`.

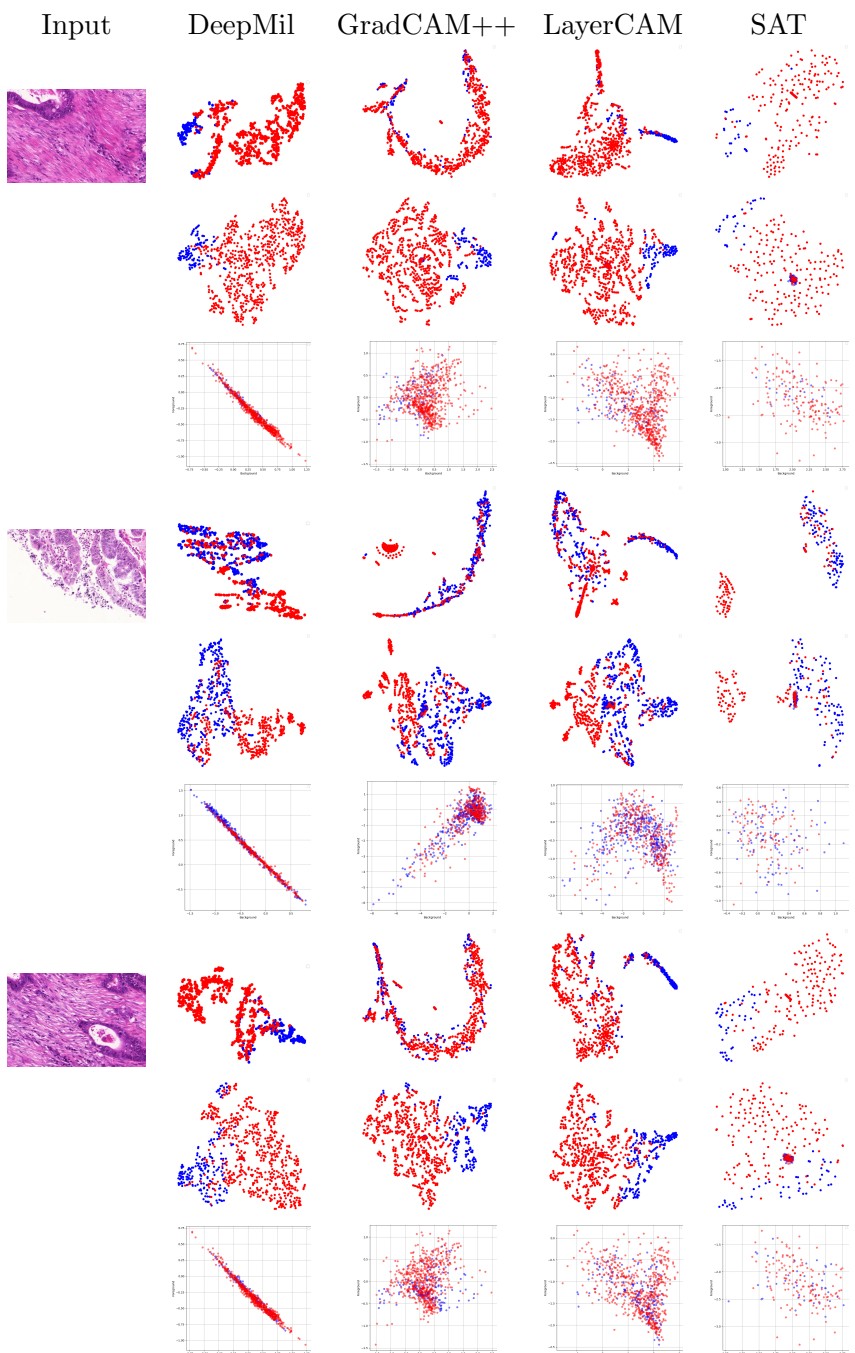

Figure 22: OOD setup (CAMELYON16 → GlaS): For the three **cancerous** images from GlaS, we display the t-SNE projection of foreground and background pixel-features of WSOL baseline methods without (1st row) and with (2nd row) PixelCAM. The 3rd row presents the logits of the pixel classifier of PixelCAM.

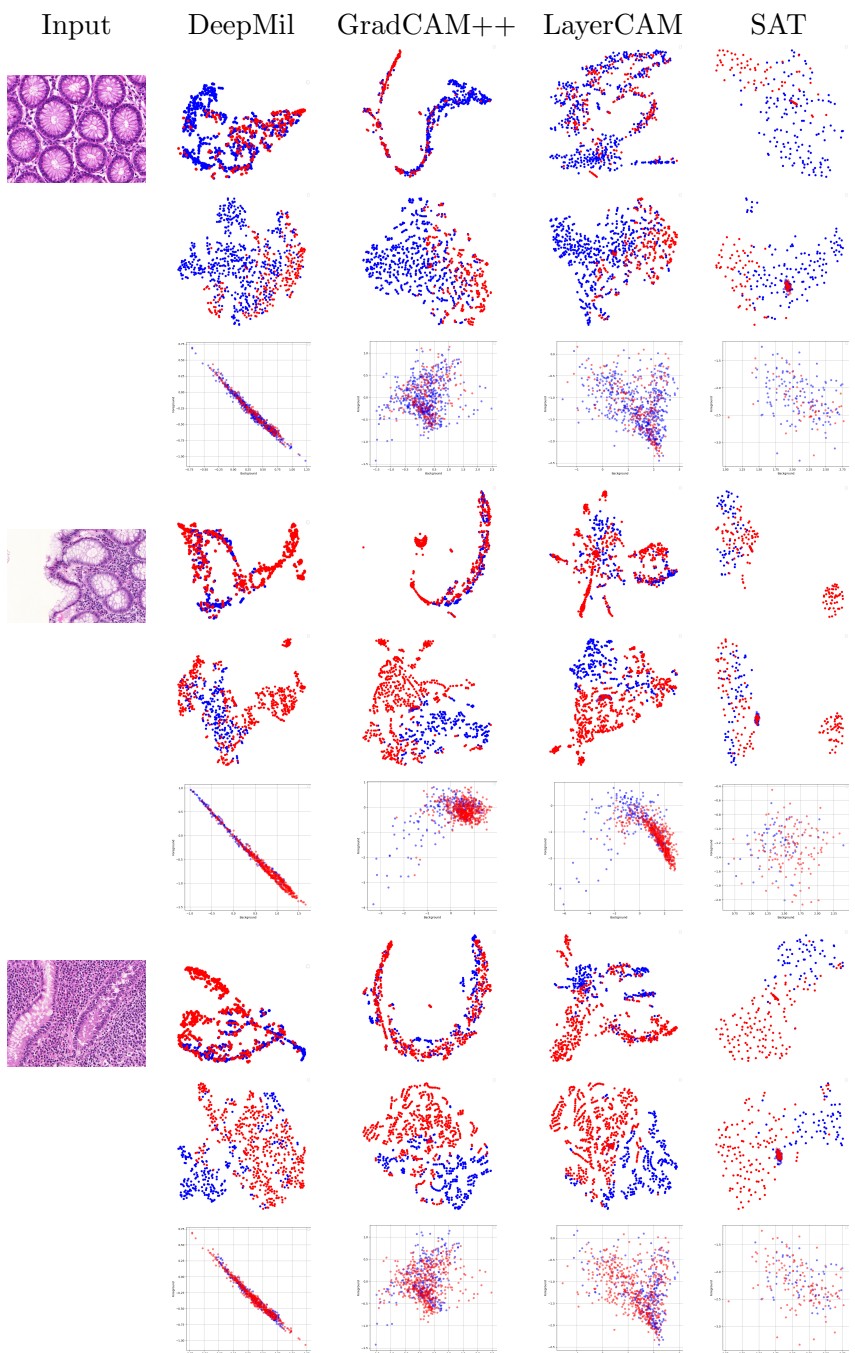

Figure 23: OOD setup (CAMELYON16 → GlaS): For the three **normal** images from GlaS, we display the t-SNE projection of foreground and background pixel-features of WSOL baseline methods without (1st row) and with (2nd row) PixelCAM. The 3rd row presents the logits of the pixel classifier of PixelCAM.

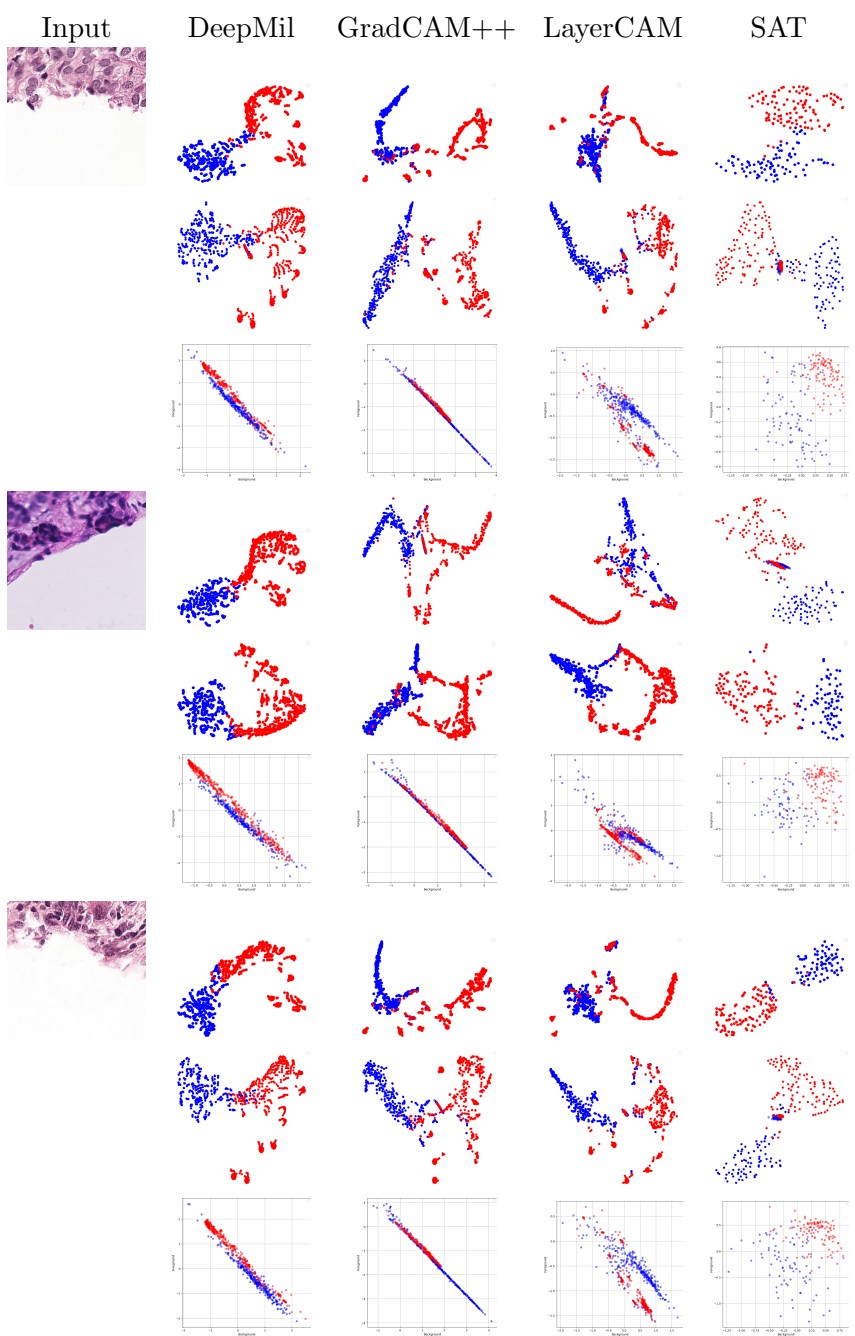

Figure 24: OOD setup (GlaS → CAMELYON16): For the three **cancerous** images from CAMELYON16, we display the t-SNE projection of foreground and background pixel-features of WSOL baseline methods without (1st row) and with (2nd row) PixelCAM. The 3rd row presents the logits of the pixel classifier of PixelCAM.

