# OpenReview forum: "PixelCAM: Pixel Class Activation Mapping for Histology Image Classification and ROI Localization"
_MIDL.io/2025/Conference — MIDL 2025 Poster_

### Official Review · Reviewer_tZNL · 2025-02-20

**Confidence:** 3
**Preliminary Rating:** 3
**Recommendation:** Poster
**Final Rating:** 4

**Summary:**

This paper presents PixelCAM, a technique for weakly supervised object localization (WSOL) in histology images, focusing on minimizing annotation costs while preserving the ability to classify and segment medical images accurately. The method uses a single model with two “heads”: an image-level classifier and a lightweight pixel classifier trained on pseudo-labels derived from a pretrained CAM-based approach. By simultaneously learning classification and pixel-level foreground vs. background separation, the approach aims to avoid common pitfalls of WSOL, such as inconsistent convergence or poor localization in challenging histology data. Experiments on GlaS (colon) and CAMELYON16 (lymph nodes) demonstrate noteworthy gains in both classification and ROI localization, with ablation studies and domain-shift experiments supporting the practicality of the method. Nonetheless, certain details—such as pseudo-label generation, architectural choices, and real-world readiness—could be elaborated further.

**Strengths:**

The paper addresses a relevant problem in medical imaging, where dense pixel annotations are highly expensive. By coupling an image classifier and a pixel classifier into a multi-task framework, the authors potentially strike a balance between labeling efficiency and localization accuracy. Empirical results show improved performance on two standard histology datasets, and it is commendable that domain-shift scenarios (where training and testing data come from different cancer types) are also evaluated. The visualizations (e.g., t-SNE plots) and various ablation studies add depth to the experimental analysis, while the paper’s structure is relatively concise and backed by clear figures. This design choice—training a compact pixel classifier with minimal overhead—may encourage broader adoption among researchers who need an interpretable solution but cannot collect large-scale pixel labels. Also availability of the code is appreciated.

**Weaknesses:**

Some statements use “significant” when reporting performance differences, but the paper does not show statistical tests or confidence intervals. This undermines the strength of claims about outperformance. Another key question relates to the pretrained WSOL model from which pseudo-labels are generated: it is unclear whether pretraining is done on the same dataset, and if so, whether that might overly bias the pseudo-labeling process. The rationale behind choosing ResNet-50 and DeiT-Tiny as backbones is also only briefly mentioned. Although the authors stress the advantage of partial annotation, one might argue that in critical settings, the highest possible accuracy outweighs the cost reduction. The method occasionally produces over-segmentation (particularly with GlaS normal cases), a phenomenon that calls for deeper investigation into thresholding or morphological constraints. Finally, further discussion on computational complexity (hardware, memory usage, etc.) is missing, and the authors do not include a section explicitly discussing method limitations or future improvements.

**Detailed Comments:**

- If the authors claim that their method provides “significant” improvements, it would be prudent to include t-tests, p-values, or confidence intervals to support these statements.
- The paper states that pseudo-labels are derived from a pretrained CAM method, but it does not specify if that pretraining occurs on the exact same data or a separate corpus. Clarifying this is essential for reproducibility.
- A short section detailing why ResNet-50 and DeiT-Tiny were chosen (e.g., parameter count, interpretability, or domain-specific performance) would strengthen the paper’s experimental rationale.
- While lowering annotation costs is commendable, the method’s final accuracy for ROI localization must meet practical clinical requirements. The paper should acknowledge that in real diagnostic workflows, small performance improvements may outweigh large gains in annotation efficiency.
- How would a pathologist incorporate PixelCAM in daily practice? For example, if the localization is still off by a margin on extremely challenging samples, is there an interactive correction step?
- On some GlaS cases it looks like large over-segmentations are happening. A comment on that would be appreciated.
- Include GPU specifications or runtime comparisons to help potential adopters understand the cost of adding a pixel classifier.
- The study omits a discussion of limitations—such as domain shifts that remain difficult, edge cases (normal tissue with subtle anomalies), or partial failures in segmentation. An outlook on prospective solutions or expansions would be welcome.
- To ensure the method's robustness in clinical settings, a discussion of strategies for managing critical false positives—such as through threshold adjustment, post-processing, or incorporating morphological constraints—could greatly enhance the practical applicability of PixelCAM.
- Introducing terms like WSOL, CAM, etc again in the main text (even if they appear in the abstract) would make the manuscript more accessible to readers.
- In the result tables, simple up/down arrows for metrics could help readers parse the results easier.
- There is another data set you might be interested in for future work: https://doi.org/10.1038/s41597-020-00608-w

**Justification Of The Final Rating:**

Thank you very much for taking so much time to answer the questions raised! I really appreciate the effort that has been made to resolve all my concerns. I am therefore changing my rating to weak accept. Nice work, I look forward to seeing where this approach goes in the future.

**Justification Of The Preliminary Rating:**

Despite showing promising results on benchmark datasets, the paper provides limited insight into how this approach would feasibly translate into real clinical workflows. Weakly supervised methods can be appealing for reduced annotation costs, but the continued need for very high accuracy in medical settings underscores potential gaps between research findings and practical deployment. In particular, the paper’s over-segmentation in certain cases and the lack of a clear plan to manage critical false positives raise doubts about immediate real-world utility.

**Questions To Address In The Rebuttal:**

Could you clarify the exact protocol for training the baseline CAM model that provides pseudo-labels, including whether it is the same dataset or external data, and how this choice affects final results?
What factors motivated the selection of ResNet-50 and DeiT-Tiny, and have you explored alternative backbones (such as EfficientNet or Swin Transformer)?
Over-segmentation appears in GlaS normal tissue. Would threshold refinements or morphological post-processing (e.g., removing small blobs) mitigate this, and have you tested such approaches?
Can you offer a brief quantitative or qualitative comparison of runtime and memory usage between PixelCAM-integrated WSOL vs. baseline methods, possibly including GPU information?
Do you see a scenario in which medical workflows might accept lower localization accuracy for reduced labeling labor, or is this method better suited as a preliminary screening that pathologists must verify?

---

> ### Author Response · Authors · 2025-03-08
> **Official comments by Authors**
>
> Thank you for your constructive comments. We appreciate that the reviewer values our proposed solution.
>
> **New tables (or figures) referenced in our response can be found in this document: new_tables.pdf in the zipped file in the rebuttal box.**
>
>
> "**If the authors claim that their method provides “significant” improvements, it would be prudent to include t-tests, p-values, or confidence intervals to support these statements.**"
>
> Thank you for your comment. Currently, we follow the same evaluation protocol as previous works in WSOL, which reports classification and localization performance using the CL and PXAP metrics [1,2,3] to assess the effectiveness of methods. However, we appreciate your suggestion and agree that incorporating statistical tests would provide additional insights. As this requires multiple runs for each baseline, which is not feasible within the current time frame, we suggest including these results in the final version of the paper for the camera-ready version. In addition, we have removed strong statements like 'significant' to describe the performance of our pixelcam method from the paper and supplementary material until these statistical tests can support that our results are significantly better.
>
> "**The paper states that pseudo-labels are derived from a pretrained CAM method, but it does not specify if that pretraining occurs on the exact same data or a separate corpus. Clarifying this is essential for reproducibility.**"
>
> PixelCAM uses a WSOL CAM-based model to obtain pseudo label for foreground and background. In our experiments, we considered training a standard classic WSOL CAM-based (DeepMIL, GradCAM++, LayerCAM, SAT) on the same dataset to obtain pixel pseudo-labels. We clarify this in the paper in section 2.
>
> However, one can pretrain the model on an external dataset. We recommend to use an external dataset with similar distribution as the source in this case. Difference in distribution will lead to poor localization and therefore, poor pseudo-labeling as shown in [2]. We compare the performance of a pretrained model in the case of using the same or external dataset for pretraining, in addition to comparing our method. We did this on GLAS and CAMELYON16 datasets. We use the other dataset as the external dataset in each case. We report the results in Tab.2 (in file new_tables.pdf). We observe that in both cases, the pretrained model yields better performance on both tasks when trained on the same dataset. Consequently, this leads to better pseudo-labeling, and therefore, a better PixelCAM.
>
> "**Selection of ResNet-50 and DeiT-Tiny**"
>
> First, we aimed to explore the performance of PixelCAM over different architectures, including both CNN-based and Transformer-based models. A previous study [1], performed a comparative analysis of various WSOL CAM-based models, considering architectures such as VGG, Inception, ResNet, and DeiT. Additionally, a recent paper [2] assessing the performance of standard WSOL CAM-based models in domain adaptation primarily used ResNet-50 and DeiT-Tiny as backbones. These architectures have been widely employed in recent studies [3], reinforcing their relevance as standard baselines. However, in our experiments, we extended this analysis by incorporating VGG-16 as a backbone for comparison with ResNet-50 (Tab.3 (in file new_tables.pdf)). Implementing and runnings experiments in this short time is challenging. However, we will be able to include the result for the swin-transformer [5] architecture in addition of DeiT as you suggest by the camera ready version deadline and also extend all the result with another backbone for CNN model such as VGG, Inception as proposed.
>
> "**While lowering annotation costs is commendable, the method’s final accuracy for ROI localization must meet practical clinical requirements. The paper should acknowledge that in real diagnostic workflows, small performance improvements may outweigh large gains in annotation efficiency.**"
>
> This is an interesting question, especially for practitioners. Although we do not have an exact idea on what is exactly acceptable for practical clinicians, we will discuss this aspect more with our expert pathologist. Based on our results, they provided an optimistic feedback. As for the trade-off between performance and annotation cost gain, we agree that sometimes, practitioners may prefer slightly higher annotation cost with good performance compared to gain in annotation cost with poor performance. However, this work considers a constant and low cost for all methods. For the study of cost-effective learning, active learning is a more suitable framework. We have a previous work that links WSOL setup and active learning with application to histology image segmentation [4]. We showed that simply densely annotating few samples in natural scene images, a model can have large performance gap compared to its counter-part WSOL method. However, for histology data, one needs more labeled samples.

---

> > ### Author Response · Authors · 2025-03-08
> > **Official comments by Authors**
> >
> > "**How would a pathologist incorporate PixelCAM in daily practice?**"
> >
> > Whole slide images (WSIs) are a typical usage case. These WSIs have high resolution (millions of pixels), and their analysis can be time-consuming for a pathologist to analyze the entire image. PixelCAM can be used as a preliminary step to scan the entire image and highlight ROIs. These regions can be inspected by a pathologist. Highlighting regions of interest can be performed using their CAMs, i.e., continuous maps where high activations indicate cancerous regions for example. However, one can threshold these maps to obtain discrete maps. One can control the threshold to adjust false positives/negatives rates.
> >
> > Considering this approach as a preliminary step, the pathologist can correct the highlighted regions. Such valuable feedback can be used differently. Our model can not directly use it. However, other methods on top of it can benefit from interactive feedback. In addition, this feedback can be stored and used to update our model offline later.
> >
> > "**On some GlaS cases it looks like large over-segmentations are happening. A comment on that would be appreciated.**"
> >
> > Our PixelCAM method performs better in terms of PxAP accuracy by detecting more ROIs than standard CAM-based methods for WSOL. However, as mentioned, PixelCAM can produce some over-segmentation, especially on the GlaS dataset for cancer images. This problem may be due to the size of the feature maps being processed. Indeed, for a CNN-based architecture such as ResNet-50 and an input image of size 3x224x224, the backbone produces feature maps of size 28x28, leading to a downsampling factor of 8. In other words, each pixel in the feature maps corresponds to 64 pixels in the original input space. After a CAM is produced by the classifier, it is interpolated. Pixels at the border of the ROIs can be extended to neighbors and produce this over-segmentation effect.
> >
> > Future works can explore using different classification architectures by considering an efficient upscaling of the features by a ratio of 2 or 4 to limit this problem. However, it will require a trade-off between accuracy, computation time, and memory consumption as we operate with more features per image. The strategy of interpolating features has been explored in an early stage of this work, and it has been discarded due to the large computational cost and memory usage, making it an impractical solution.
> >
> > "**Include GPU specifications or runtime comparisons to help potential adopters understand the cost of adding a pixel classifier.**"
> >
> > The computation overhead of the pixel classifier in PixelCAM is minimal in our context, which makes it suitable for analyzing a Whole Slide Image (WSI) that contains millions of pixels. From a memory point of view, the impact is negligible as the number of additional parameters depends essentially on feature size. For instance, in the ResNet50-CNN architecture, the feature vector size is equal to 2048, and the number of classes is equal to two (FG and BG). The pixel classifier adds only 4,098 parameters to the CNN-based architecture, which already contains more than 23.5M parameters. We can conclude the same for transformer-based architecture ($>$ 5.5M parameters) where we only add 386 parameters. For the inference computational cost, our pixel classifier is particularly advantageous, especially compared to WSOL gradient-based models as shown in the following table (Tab.1 (in file new_tables.pdf)). Therefore, our model does not incur serious computation cost allowing fast training and inference. The time efficiency, combined with the robustness of PixelCAM, makes it a key advantage for deployment in practical scenarios where WSI image sizes can be extremely large [1].  These details have been added to the supplementary materials in Appendix F.

---

> > ### Author Response · Authors · 2025-03-08
> > **Official comments by Authors**
> >
> > "**The study omits a discussion of limitations—such as domain shifts that remain difficult, edge cases (normal tissue with subtle anomalies), or partial failures in segmentation. An outlook on prospective solutions or expansions would be welcome**"
> >
> > Indeed, deep learning models are known to face issues when processing unseen target datasets. Our PixelCAM method also faces this problem, but comparably less than standard WSOL CAM-based models. As mentioned in the paper, especially in the visualization, we can observe that some ROIs are not highlighted. A common explanation comes from the distribution between the features generated between the source and target dataset. To alleviate this problem, several domain adaptation techniques can be leveraged to deal with domain shift. These methods are usually applied to image classifiers. When the domain discrepancy or shift becomes more severe between source and target data, some works have focused on gradual or multi-step domain adaptation. However, since we introduce a pixel classifier, we can draw inspiration from them to improve WSOL methods when dealing with domain shift.
> >
> > Domain adaptation methods have been proposed at the image level, using information contained in the image classifier with contrastive learning or aligning distributions to a common representation space by using classifier information. Our new architecture can pave the way for new research by operating at the pixel level.
> >
> >
> > "**To ensure the method's robustness in clinical settings, a discussion of strategies for managing critical false positives—such as through threshold adjustment, post-processing, or incorporating morphological constraints—could greatly enhance the practical applicability of PixelCAM.**"
> >
> > Our method, like other WSOL methods, yields activation maps to highlight potential regions of interest on images. One can threshold these maps (probability maps in our model) depending on the accepted false positive/negatives rates to produce segmentation maps. These maps can also be fed to other post-processing techniques such as Conditional Random Field (CRF) to better align the edges and even expand the maps under constraints.
> >
> > "**Introducing terms like WSOL, CAM, etc again in the main text (even if they appear in the abstract) would make the manuscript more accessible to readers.**"
> >
> > Agreed. We have redefined these acronyms in the introduction to help the reader.
> >
> > "**In the result tables, simple up/down arrows for metrics could help readers parse the results more easily.**"
> >
> > Agreed. We added up and down arrows on metrics in all tables to make it easier for the readers.
> >
> > "**There is another data set you might be interested in for future work:**"
> >
> > Thanks for the suggestion. We are presently working on the same protocol as defined in previous papers for WSOL for histology. We will investigate this dataset for potential future work. We are interested in datasets that have both global image class and pixel-wise annotations.
> >
> > References:
> >
> > [1] J. Rony, S. Belharbi, J. Dolz, I. Ben Ayed, L.e McCaffrey, E. Granger "Deep Weakly-Supervised Learning Methods for Classification and Localization in Histology Images: A Survey", MELBA 2023
> >
> > [2] A. Guichemerre, S. Belharbi, Tsiry Mayet, S. Murtaza, P. Shamsolmoali, L. McCaffrey, E. Granger "Source-Free Domain Adaptation of Weakly-Supervised Object Localization Models for Histology", CVPRw 2024
> >
> > [3] S. Murtaza, S. Belharbi, M. Pedersoli, E. Granger "A Realistic Protocol for Evaluation of Weakly Supervised Object Localization", WACV 2025
> >
> > [4] S. Belharbi , I. Ben Ayed , L. McCaffrey, E. Granger. "Deep active learning for joint classification \& segmentation with weak annotator". WACV 2021.
> >
> > [5] Z. Liu, Y. Lin, Y. Cao, H. Hu, Y. Wei, Z. Zhang, S. Lin, B. Guo "Swin Transformer: Hierarchical Vision Transformer using Shifted Windows". ICCV 2021.

---

> > > ### Comment · Reviewer_tZNL · 2025-03-12
> > >
> > > Thank you very much for taking so much time to answer the questions raised! I really appreciate the effort that has been made to resolve all my concerns. I am therefore changing my rating to weak accept. Nice work, I look forward to seeing where this approach goes in the future.

---

> > ### Author Response · Authors · 2025-03-12
> > **Official comments by Authors**
> >
> > Dear Reviewer "tZNL",
> >
> > We sincerely thank you for considering our rebuttal, and valuable effort in reviewing our work. We are glad that our response was satisfactory for you. We are currently working to finish the ongoing mentioned simulations that will be included in the final version.
> >
> > Thank you again,
> >
> > Best,
> >
> > The authors

---

### Official Review · Reviewer_evkC · 2025-02-20

**Confidence:** 4
**Preliminary Rating:** 4
**Recommendation:** Poster
**Final Rating:** 4

**Summary:**

This paper introduces PixelCAM, a novel multi-task approach for weakly supervised object localization (WSOL) in histology images. By integrating a lightweight pixel-wise classifier into the shared feature space of a deep network, PixelCAM learns to explicitly separate foreground and background regions using pseudo-labels extracted from a pretrained CAM model. This simultaneous optimization for both image classification and ROI localization addresses the asynchronous convergence issue that often plagues standard WSOL methods. Extensive experiments on the GlaS and CAMELYON16 datasets demonstrate significant improvements in localization accuracy (measured via PxAP) and classification performance, as well as enhanced robustness to domain shifts, as evidenced by improved pixel feature separability. The method’s compatibility with both CNN- and transformer-based architectures further underlines its versatility and practical value in computational pathology

**Strengths:**

The paper presents a well-motivated and innovative solution to a longstanding challenge in WSOL for histology images. One major strength is the introduction of a pixel-wise classifier that operates in the feature space, which not only improves ROI localization but also enhances global image classification. The multi-task framework ensures that both tasks are optimized concurrently, thereby mitigating issues related to asynchronous convergence. Extensive experiments, including detailed ablation studies on classifier depth, pixel sampling techniques, and hyperparameter sensitivity, provide strong empirical support for the proposed approach. Moreover, the method’s ability to handle out-of-distribution data and its easy integration into different network architectures make it a valuable contribution to the field.

**Weaknesses:**

1. The reliance on pseudo-labels from a pretrained WSOL CAM model means that the overall performance of PixelCAM is closely tied to the quality of these initial labels; inaccuracies in the pseudo-labels could adversely affect both classification and localization outcomes.
2. While the ablation studies are comprehensive, the paper could benefit from a deeper discussion on the computational overhead introduced by the additional pixel classifier, particularly in transformer-based settings. Some comparisons with fully supervised localization methods or alternative multi-task approaches would also strengthen the empirical validation.
3. Additional error analysis under extreme domain shift scenarios could help clarify the method’s limitations in challenging real-world environments

**Detailed Comments:**

The paper is well-structured and clearly written, providing a comprehensive description of the PixelCAM framework along with a solid experimental evaluation on standard histology datasets. The integration of a pixel classifier into the feature extraction pipeline is both innovative and effective, as demonstrated by significant gains in localization accuracy and classification performance. The detailed ablation studies offer valuable insights into the impact of various design choices, such as the classifier’s depth and the pixel sampling strategy. However, the dependence on pseudo-label quality and a lack of extensive comparisons with other state-of-the-art WSOL or fully supervised methods leave some questions unanswered. Additional discussion regarding the computational complexity and potential failure modes under severe domain shifts would further improve the paper.

**Justification Of The Final Rating:**

The authors’ rebuttal effectively addresses my key concerns. They convincingly demonstrate that the pseudo-label strategy, based on stochastic sampling from CAMs, minimizes noise and supports gradual learning of discriminative features, while the additional pixel classifier incurs negligible computational overhead even in transformer-based architectures. The comparisons with recent methods and the provided supplementary analyses underline PixelCAM’s competitive performance and robustness under domain shift, despite some acknowledged limitations. Because of the clarifications and the extensive experimental results provided, I vote for weak accept.

**Justification Of The Preliminary Rating:**

The paper introduces a compelling and effective approach to improving WSOL for histology image analysis by incorporating a pixel-wise classifier into a multi-task learning framework. The experimental results on the GlaS and CAMELYON16 datasets, along with extensive ablation studies, provide strong evidence of its benefits in terms of both classification and localization performance, as well as enhanced robustness to domain shifts. However, the method’s dependency on pseudo-label quality and some aspects of its computational complexity are not fully addressed. These issues, along with the need for additional comparisons with other state-of-the-art methods, prevent a stronger endorsement. Overall, the contribution is valuable and advances the field, though certain aspects could be clarified or improved upon in future work

**Questions To Address In The Rebuttal:**

1. How sensitive is PixelCAM to the quality of the pseudo-labels extracted from the pretrained WSOL model, and what measures are in place to mitigate potential errors from inaccurate pseudo-labels?
2. Could the authors provide more detailed insights into the computational overhead introduced by the pixel classifier, particularly when integrated into transformer-based architectures?
3. How does PixelCAM compare with other contemporary multi-task WSOL approaches or fully supervised localization methods in terms of both performance and computational efficiency?
4. What are the observed failure modes under extreme domain shift scenarios, and how might these be addressed in future work?

---

> ### Author Response · Authors · 2025-03-08
> **Official comments by Authors**
>
> Thank you for your constructive comments. We particularly appreciate that the reviewer emphasized the key impacts of PixelCAM.
>
> **New tables (or figures) referenced in our response can be found in this document: new_tables.pdf in the zipped file in the rebuttal box.**
>
> "**How to mitigate the impact of inaccurate pseudo labels during the training of PixelCAM**"
>
> Using pseudo-labels from CAMs in our work is mainly motivated by recent works [4, 5, 6]. They showed that CAMs can be used as a source of noisy pseudo-labels. But most importantly, the work in [4] showed that stochastically sampling locations as pseudo-labels based on CAMs can mitigate the noise, compared to when using standard fixed pseudo-labels. The intuition behind this empirical result is that randomly sampling the most reliable locations reduces the model's overfitting to incorrect pseudo-labels compared to fixed pseudo-labels. In addition, unlike learning with fixed pseudo-labels, random sampling allows the model to gradually learn discriminative vs BG regions. This allows for more accurate regions to emerge gradually through learning. In addition, fitting pseudo-labels is weighted using the coefficient ${\lambda}$ in Eq.3 to control how much we trust the pseudo-label.
>
> "**How sensitive is PixelCAM to the quality of the pseudo-labels extracted from the pretrained WSOL model?**"
>
> Figure 1 (in file new_tables.pdf)  illustrates the evolution of true positive and negative rates on the percentage of selected pixels, considering both FG and BG for GLAS and only FG for CAMELYON16. We observe that high-intensity pixels mostly correspond to true positives, justifying their prioritization in selection. However, beyond a certain threshold,the  true positive rate decreases, introducing incorrect pseudo-labels. Our PixelCAM method relies on pixel-level supervision using pseudo-labels from a pretrained WSOL-based CAM model. The analysis of the impact of the number of selected pixels (Section H (in file WSOL_Histology_Image_Pixel_CAM.pdf)) shows that increasing this selection up to 20 slightly reduces localization performance on GLAS (-0.7), while on CAMELYON16, performance remains stable or even improves. It is explained by the high accuracy of true positives, exceeding 80\% on GLAS and 65\% on CAMELYON16 for a selection between 1 and 20. However, beyond 40\%-50\%, we note an important decrease in the true positive rate, which can negatively impact the training of PixelCAM.
>
> "**Computational overhead introduced by our pixel classifier**"
>
> The computation overhead of the pixel classifier in PixelCAM is minimal in our context. It is suitable for analyzing whole slide images (WSIs) that contain millions of pixels. From a memory point of view, the impact is negligible since the number of supplementary parameters depends essentially on feature size. For instance, in the ResNet50-based architecture, the feature size is equal to 2048, and the number of classes is equal to two (FG and BG). The pixel classifier adds only 4,098 parameters to the CNN-based architecture, which already contains more than 23.5M parameters. The same conclusion applies to the transformer-based architecture ($>$ 5.5M parameters), where we add 386 parameters only.
>
> For the inference computational cost, our pixel classifier is particularly advantageous, especially compared to WSOL gradient-based models for WSOL (see Tab.1 in file new_tables.pdf). Therefore, our model incurs negligible computation overhead, allowing for fast training and inference. The time efficiency, combined with the robustness of PixelCAM, makes it a key advantage for deployment in practical scenarios where WSI image sizes can be extremely large [1].  These details have been added to the supplementary materials in Appendix F.
>
> "**How does PixelCAM compare with other contemporary multi-task WSOL approaches or fully supervised localization methods in terms of both performance and computational efficiency?**"
>
> We compared our pixelcam to the NEGEV method, which is a multi-task method. Our method shows competitive results in standard WSOL with fewer computations. As for fully supervised methods, we provided a comparison to the U-Net method, showing that there is still a large gap to be covered by WSOL methods. We are currently running experiments on a more recent WSOL method [7], and we will provide the results in the discussion soon.

---

> > ### Author Response · Authors · 2025-03-08
> > **Official comments by Authors**
> >
> > "**What are the observed failure modes under extreme domain shift scenarios, and how might these be addressed in future work**"
> >
> > WSOL methods are known to underperform when applied to new datasets because of domain shift [2]. While our model is somewhat affected by this problem, we observed better results in terms of PxAP (localization) and CL (classification). Table 2 (in file WSOL_Histology_Image_Pixel_CAM.pdf) indicates that pixelcam provides some robustness to distribution change.  However, the model may still struggles to maintain the same level of performance as on the source, especially in the case of extreme shift (from GlaS to CAMELYON16). This can be mainly explained by the small number of training data in the source, but also due to the wide variety of tissues in the target dataset (CAMELYON16), leading to a lower localization performance.
> >
> > To address this issue, several techniques in domain adaptation strategies could be explored. Many existing approaches rely on feature alignment techniques such as contrastive learning or distribution alignment to reduce the domain shift [2,3]. Our new pixel classifier can serve exactly as the image classifier by adapting such techniques at the pixel level. Additionally, in domain adaptation context, most of the techniques use clustering techniques to refine pseudo labels. Since our model produces more discriminative features (as shown in Table. 3 and 4 (in file WSOL_Histology_Image_Pixel_CAM.pdf) ), PixelCAM can improve clustering effectiveness leading to more reliable pixel pseudo-label for adaptation approaches. We believe that the introduction of our PixelCAM model can open the doors to new opportunities in domain adaptation in WSOL task  by leveraging a novel source of information based on the pixel classifier.
> >
> > References:
> >
> > [1] J. Rony, S. Belharbi, J. Dolz, I. Ben Ayed, L.e McCaffrey, E. Granger "Deep Weakly-Supervised Learning Methods for Classification and Localization in Histology Images: A Survey", MELBA 2023.
> >
> > [2] A. Guichemerre, S. Belharbi, T. Mayet, S. Murtaza, P. Shamsolmoali, L. McCaffrey, E. Granger. "Source-Free Domain Adaptation of Weakly-Supervised Object Localization Models for Histology", CVPRw 2024.
> >
> > [3] J. Li, Z. Yu, Z. Du, L. Zhu, H. Tao Shen "A comprehensive survey on source-free domain adaptation", TPAMI 2024.
> >
> > [4] S. Belharbi, M. Pedersoli, I. Ben Ayed, L. McCaffrey, and E. Granger. "Negative evidence matters in interpretable histology image classification". In MIDL 2022.
> >
> > [5] S. Belharbi, A. Sarraf, M. Pedersoli, I. Ben Ayed, L. McCaffrey, and E. Granger. "F-CAM: Full resolution class activation maps via guided parametric upscaling". In WACV 2022.
> >
> > [6] S. Belharbi, S. Murtaza , M. Pedersoli, I. Ben Ayed, L. McCaffrey, E. Granger. Colo-cam: "Class activation mapping for object co-localization in weakly-labeled unconstrained videos". Pattern Recognition, 2025.
> >
> > [7] S. Murtaza, M. Pedersoli, A. Sarraf, E. Granger "Leveraging Transformers for Weakly Supervised Object Localization in Unconstrained Videos". ANNPR 2024

---

> > > ### Author Response · Authors · 2025-03-14
> > > **Official comments by Authors**
> > >
> > > Dear reviewer "evkC",
> > >
> > > We sincerely appreciate the time and effort you have dedicated to reviewing our work. We hope that you have received our rebuttal and that it addresses your concerns.
> > >
> > > We remain available to address any questions or provide clarification regarding our rebuttal.
> > >
> > > Thank you again,
> > >
> > > Best,
> > >
> > > The authors

---

### Official Review · Reviewer_nHGf · 2025-02-22

**Confidence:** 5
**Preliminary Rating:** 4
**Recommendation:** Oral
**Final Rating:** 5

**Summary:**

This work tackles weakly supervised object localization (WSOL). The authors propose a multitask single-step WSOL method named PixelCAM for histology image analysis. PixelCAM consists of encode, image classifier and localiser (pixel classifier). They train these three components simultaneously, where class labels are used for training of classifier and pseudo-annotation-labels of background and foreground given by a pre-trained WSOL model (CAM based method) are used for training of localiser. In experimental evaluations, the authors evaluate the proposed method by using several CAM methods as pretrained WSOL methods for two datasets.

**Strengths:**

- The authors challenge the localization problem without annotations via a weakly supervised manner.
- They evaluate the proposed method using several CAM based weakly supervised object localisation (WSOL).
- They presented comparison with the baseline method and evaluations for domain shift problem.
- It is interesting that background/foreground pseudo labels work well even CAM-based approach.

**Weaknesses:**

- Technical novelty is unclear since many CAM-based WSOL exist. A joint learning of classifier and localiser is normal approach, not special.
- The mechanism why background/foreground pseudo label can mitigate domain shift issue is unclear and unconvincing.

**Detailed Comments:**

- In section2, mathematical notation usages are bumpy. Please use standard mathematical notations.
- Presenting what is the main difference among the proposed and existing WSOL by suggesting theoretical mechanism might be more convincing.
- Generally, CAM-based pseudo labels apt to be inaccurate and can lead to insufficient localisation performance. If there are several assumptions or theoretical reasons, these should be clarified.

**Justification Of The Final Rating:**

The authors' feedback includes unclear claim but the main idea and approach should be interesting for MIDL community. Therefore, I think this work is welcome for  a poster presentation (even though my preliminary rating was oral), where many attendees can discuss the unclear points with the presenter.

**Justification Of The Preliminary Rating:**

The submission have several strength listed above but it need explanation for justifying their main idea. I have interested in the reason why background/foreground pseudo labels work well even CAM-based approach.

**Questions To Address In The Rebuttal:**

In CAM-based WSOL, inaccurate pseudo labels leads to unwelcome trained model without generalisation ability. How to improve this point is unclear in the current descriptions.

Page 4, the authors describe "Most importantly, introducing domain shift over the input image will lead to further feature confusion and failure
of subsequent modules to localize and classify. To mitigate this issue, our pixel-classifier f is used to create well-separated features between foreground/background in the pixel-feature space." However, why? Theoretical explanation is welcome.

**Special Issue:**

No

---

> ### Author Response · Authors · 2025-03-08
> **Official Comment by Authors**
>
> Thank you for your constructive comments. We appreciate that you highlighted the advantage of introducing the pixel classifier to improve PxAP accuracy on both source and target datasets.
>
> **New tables (or figures) referenced in our response can be found in this document: new_tables.pdf in the zipped file in the rebuttal box.**
>
> "**Technical novelty is unclear since many CAM-based WSOL exist.**"
>
> State-of-the-art single-step models rely on the classifier prediction to localize but struggle with less salient objects, such as those in histology images [1]. The other more successful category, two-step models, also have limitations. Some methods use separated architectures, leading to independent decisions, which are less suitable for medical applications. Other models use a decoder, which increases computational cost in addition to being limited since they freeze the encoder to preserve classification accuracy. This restriction constrains feature representation, impacting the decoder's effectiveness.
>
> In contrast, our method pixelcam introduces a WSOL architecture that directly operates on the pixel features, explicitly constraining the model to learn discriminative pixel representations. Explicitly building discriminative pixel features is a distinctive aspect of our method compared to previous WSOL methods, which is done indirectly through global image class. By doing so, we allow better distinction between discriminative regions and noise through the pixel class anchors: foreground and background. This increases the model's robustness to out-of-distribution as the feature extractor learns to build features that match these class anchors. On the other hand, standard WSOL architecture leverages only global image class, leaving pixels without direct supervision. This leads to poor localization and extreme vulnerability to domain shift, as shown in [2]. Our design of this architecture is mainly motivated by the very limited performance of WSOL methods when dealing with domain shift [2]. Our method is meant to yield competitive localization performance while being robust to domain shift. This has been shown empirically in this paper in an extreme domain shift setup where a domain is modeled by an organ (colon and breast tissue).
>
> We believe that introducing our pixel-classifier into a  WSOL method will pave the way for applying common domain adaptation methods to better deal with domain shift in WSOL scenarios. Standard WSOL methods lack this aspect. Domain adaptation methods are commonly applied with standard image classifiers. We can easily draw inspiration from these methods and adapt them to pixel classifiers, mitigating domain shift and learning robust WSOL models.
>
> Additionally, in previous WSOL approaches, the classification and localization tasks are antagonistic [1] -- improving one task will negatively impact the performance of the other task.
>
> "**Please use standard mathematical notations.**"
>
> Thank you for this comment. We have revised it and attempted to clarify. Can you please indicate which parts remain unclear so that we can further improve the notation?
>
> "**How to mitigate the impact of inaccurate pseudo labels?**"
>
> Using pseudo-labels from CAMs in our work is mainly motivated by recent works [3, 4, 5]. They showed that CAMs can be used as a source of noisy pseudo-labels. But most importantly, the work in [3] showed that stochastically sampling locations as pseudo-labels based on CAM's activation can mitigate the noise, compared to when using standard fixed pseudo-labels. The intuition behind this empirical result is that randomly sampling the most reliable locations reduces the model's overfitting to wrong pseudo-labels compared to fixed pseudo-labels. In addition, unlike learning with fixed pseudo-labels, random sampling allows the model to gradually learn discriminative vs background regions. This allows for more accurate regions to emerge gradually through learning. In addition, fitting pseudo-labels is weighted using a coefficient ${\lambda}$ in Eq.3 to control how much we trust the pseudo-label.

---

> > ### Author Response · Authors · 2025-03-08
> > **Official comments by Authors**
> >
> > "**Why background/foreground pseudo label mitigate domain shift**"
> >
> > Our primary goal is to design a robust WSOL method to domain shift. Since standard WSOL methods do not provide an explicit supervision at pixel-level where features are produced to perform both localization and image classification, pixel-features can be poor. This can lead to poor localization/classification. In addition, this makes pixel-features extremely vulnerable to domain shift [2], as input shift can lead to large deviation in feature space which can break both subsequent tasks, localization and classification.
> >
> > To combat the vulnerability to domain shift, our intuition is to use a pixel-classifier. It is typically modeled by a set of class anchors (following domain adaptation terminology) where each anchor represents its class, and it can be seen as an expected class representation or a reference. This pushes the encoder (feature extractor) to learn to build features that matches these anchors. When presented with a shift in images, this encoder is expected to produce features close to the anchors, and therefore, it reduces the impact of the input shift at feature space leading to robust localization and image classification. On the other hand, standard WSOL can be very vulnerable to shift since they are not equipped with pixel-feature reference. Our empirical results showed that our method can yield robust localization and image classification performance under extreme domain shift (between organs), compared to standard WSOL methods. In addition, we showed that our pixel-features are well separated compared to other methods suggesting that pixel-features of these methods can be easily corrupted when introducing domain shift.
> >
> > We are currently conducting another simulation to show this fact about our method. In this simulation, we control the input shift, and measure the pixel-feature shift in our method, compared to other methods.
> >
> > Building a pixel-classifier requires pixel pseudo-labels. Since in WSOL, we deal with single class per-image in a multi-class setup, we converted the problem into foreground/background pixel classification problem. We ensure that the foreground pseudo-label comes from the CAM of the true class of the image. So, it is not the use of foreground/background that makes our model robust to domain shift, it is the discriminative classifier that learns on top of them.
> >
> > Note that when dealing with multi-label problem where several classes can occur in an image, our method can be easily adapted by using a multi-label pixel-classifier.
> >
> > References:
> >
> > [1] J. Rony, S. Belharbi, J. Dolz, I. Ben Ayed, L.e McCaffrey, E. Granger "Deep Weakly-Supervised Learning Methods for Classification and Localization in Histology Images: A Survey", MELBA 2023.
> >
> > [2] A. Guichemerre, S. Belharbi, T. Mayet, S. Murtaza, P. Shamsolmoali, L. McCaffrey, E. Granger "Source-Free Domain Adaptation of Weakly-Supervised Object Localization Models for Histology", CVPRw, 2024.
> >
> > [3] S. Belharbi, M. Pedersoli, I. Ben Ayed, L. McCaffrey, and E. Granger. "Negative evidence matters in interpretable histology image classification". In MIDL, 2022.
> >
> > [4] S. Belharbi, A. Sarraf, M. Pedersoli, I. Ben Ayed, L. McCaffrey, and E. Granger. "F-CAM: Full resolution class activation maps via guided parametric upscaling". In WACV, 2022.
> >
> > [5] S. Belharbi,S. Murtaza, M. Pedersoli , I. Ben Ayed, L. McCaffrey, E. Granger. Colo-cam: "Class activation mapping for object co-localization in weakly-labeled unconstrained videos". Pattern Recognition, 2025.

---

> > > ### Author Response · Authors · 2025-03-14
> > > **Official comments by Authors**
> > >
> > > Dear reviewer "nHGf",
> > >
> > > We sincerely appreciate the time and effort you have dedicated to reviewing our work. We hope that you have received our rebuttal and that it addresses your concerns.
> > >
> > > We remain available to address any questions or provide clarification regarding our rebuttal.
> > >
> > > Thank you again,
> > >
> > > Best,
> > >
> > > The authors

---

> ### Comment · Reviewer_nHGf · 2025-03-15
>
> Thank you for the feedback.
>
> >[Technical novelty]
> In the feedback, the authors explain their explanations about the technical novelty. These explanations should be well written in a manuscript. However, the current explanations are also still unconvincing. More detailed concepts or mechanisms with slid definitions are welcome in the methodology part of the manuscript. The ablation study is also required.
>
> For pixel-wise segmentation, the Grad-CAM-based WSOL survey is missing important previous works. Pixel-wise annotations are very tough work. So, bounding-box subset-object annotations might be studied even in WSI segmentation. For example, in the polyp segmentation in colonoscopy images, the following work [1] achieved the segmentation with weak bounding-box annotations, where the work proposed a multi-scale fusion of improved Grad-CAM. Moreover, the other work [2] also reported the relation between Grad-CAM, Grad-CAM++ and Positive-gradient Grad-CAM. From the experimental results in [1], where pseudo label is completely meaningless even though the trained model has high generalization ability, I can not understand why the proposed method can improve activation maps and model learning.
>
> [1] H Itoh, et al., Positive-Gradient Weighted Object Activation Mapping: Visual Explanation of Object Detector Towards Precise Colorectal-Polyp Localisation (2022) International Journal of Computer Assisted Radiology and Surgery
>
> [2] M. A. Lerma & M Lucas, Grad-CAM++ is Equivalent to Grad-CAM With Positive Gradients (2022) 24th Irish Machine Vision and Image Processing Conference
>
> At least, compared with existing approaches, the differences and advantages of the proposed method should be addressed in the manuscript.
>
>
> >[Mathematical notation]
> >Thank you for this comment. We have revised it and attempted to clarify. Can you please indicate which parts remain unclear so that we can further improve the notation?
>
> I am just a reviewer, not your supervisor or co-author. I strongly recommend going back to your undergraduate text of mathematics.
>
> For future work, I will give you some comments. Generally, a Roman italic is used for a scalar value or function. A bold italic is used for a vector or matrix even though it depends on mathematical styles. For a tensor, different font styles from these two are used. However, the authors do not. For example, the authors use a bold style for the cross entropy H, but it should be a function that returns a scalar value. As another example, the authors use bold S and F for tensor representations. Small and capital letters are generally used for vector and matrix notations even though the authors represent a d-dimensional vector as \bm{F}_p. Even for the notation of a set, the authors use inconsistent manner such that \mathbb{D} and \bm{\theta} are both sets. Furthermore, a \bm{theta} is written in bold, but a set should be given by a roman style in general. Why should it be a bold style? Moreover, the authors used | | notation for dimension. I think the authors do not understand the difference between cardinality and dimension. Generally, | | notation is used for cardinality. It is known that the cardinality of a subset of all the real numbers in R is equal to the cardinality of R, and is aleph 1. In addition, the cardinality of a subset \Omega \subset  R^2 also equals aleph 1. So, their notation does not show the size of dimensions of a two-dimensional array. The authors do not understand the direct product of two vector spaces for defining a two-dimensional Euclidean space, which they want to define.
>
> As a result, the current writing mimics mathematical style but is not based on the mathematical basics. If the authors want to use special notations due to some necessity, they should define all the mathematical notations in a manuscript. Without it, a manuscript does not present its repeatability.

---

### Author Response · Authors · 2025-03-08
**Official Comment by Authors**

We thank all the reviewers for their valuable insights and constructive feedback on our paper. We are encouraged by their positive comments, in particular:

  * It is interesting that background/foreground pseudo labels work well even with the CAM-based approach. (Reviewer nHGf)
  * The method’s ability to handle out-of-distribution data and its easy integration into different network architectures make it a valuable contribution to the field. (Reviewer evkC)
  * This design choice -- training a compact pixel classifier with minimal overhead -- may encourage broader adoption among researchers. (Reviewer tZNL)

Some concerns were raised by multiple reviewers, and we paid particular attention to these points, in particular, the impact of noisy labels on the training stage and the computational overhead for real-world deployment. Below, we address the major concerns and provide further clarifications. Modifications in the main paper and supplementary material are indicated in red color.

---

### Author Rebuttal · Authors · 2025-03-08

**Rebuttal:**

This zipped file contains the following material:
1. The updated paper (**WSOL_Histology_Image_Pixel_CAM.pdf**) where we highlight in red the updated parts.
2. The file **new_tables.pdf** which contains new tables/figures used in our response.

**Supporting Material:**

/attachment/e9c8508e430b1c2760791d2b67b235cb68e28d28.zip

---

### Meta-Review · Area_Chair_ys31 · 2025-03-21

**Recommendation:** Accept (Poster)
**Confidence:** 4

**Metareview:**

- Overall Evaluation

This paper proposed a new method to generate the class activation map for weakly supervised learning on WSI processing. The main idea is to build an encoder-based framework to predict CAM based on multiple-head supervision. Experiments demonstrated the effectiveness of the proposed method.

- Strength

Weakly-supervised learning for WSI is an important topic in MIDL community since the labeling of such a high-res image can be quite expensive. This paper proposed a simple pipeline for this task based on CAM, which has been widely demonstrated the sound performance. The organization and presentation of the manuscript are also promising, making it easy for me to follow.

- Weakness

As pointed by the reviewers, the main concerns of this submission are the technical novelty and the analysis. I agree with the comments that CAM-like methods have been widely investigated in previous works. Current literature review and comparisons are not sufficient to fully support the claims from the authors. Besides, it is also expected to see more statisitical analysis to evaluate the robustness of the proposed method. However, current version is still a good work and worth the acceptance to MIDL.